# NeSF: Neural Semantic Fields
# for Generalizable Semantic Segmentation of 3D Scenes

**Suhani Vora**[*]                                                                    *svora@google.com*
*Google Research*

**Noha Radwan**[*]                                                                   *noharadwan@google.com*
*Google Research*

**Klaus Greff**                                                                       *klausg@google.com*
*Google Research*

**Henning Meyer**                                                                    *tutmann@google.com*
*Google Research*

**Kyle Genova**                                                                       *kgenova@google.com*
*Google Research*

**Mehdi S. M. Sajjadi**                                                              *msajjadi@google.com*
*Google Research*

**Etienne Pot**                                                                       *epot@google.com*
*Google Research*

**Andrea Tagliasacchi**                                                              *atagliasacchi@google.com*
*Google Research, Simon Fraser University*

**Daniel Duckworth**                                                                 *duckworthd@google.com*
*Google Research*

**Reviewed on OpenReview:** *https://openreview.net/forum?id=ggPhsYCsm9*

## Abstract

We present *NeSF*, a method for producing 3D semantic fields from posed RGB images alone. In place of classical 3D representations, our method builds on recent work in neural fields wherein 3D structure is captured by point-wise functions. We leverage this methodology to recover 3D density fields upon which we then train a 3D semantic segmentation model supervised by posed 2D semantic maps. Despite being trained on 2D signals alone, our method is able to generate 3D-consistent semantic maps from novel camera poses and can be queried at arbitrary 3D points. Notably, NeSF is compatible with any method producing a density field. Our empirical analysis demonstrates comparable quality to competitive 2D and 3D semantic segmentation baselines on complex, realistically-rendered scenes and significantly outperforms a comparable neural radiance field-based method on a series of tasks requiring 3D reasoning. Our method is the first to learn semantics by recognizing patterns in the *geometry* stored within a 3D neural field representation. NeSF is trained using purely 2D signals and requires as few as one labeled image per-scene at train time. No semantic input is required for inference on novel scenes.

---

[*]Denotes equal contribution.

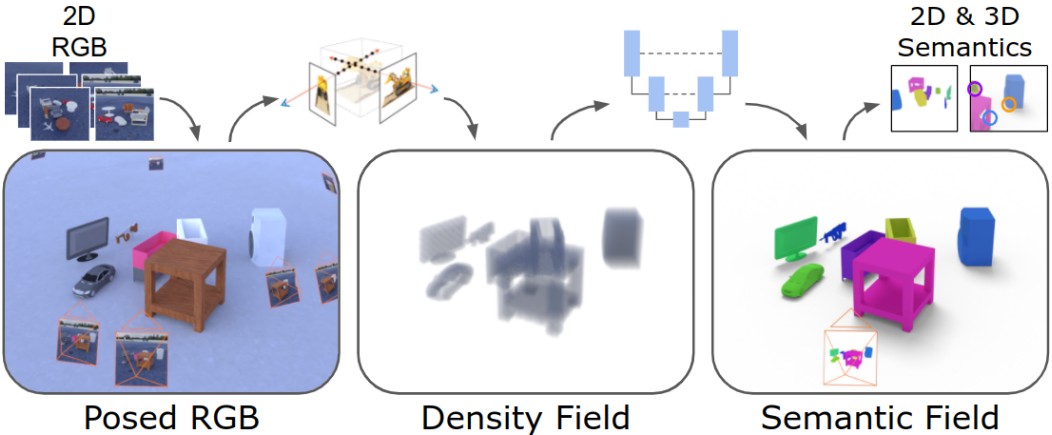

Figure 1: **Overview** – We train our method on collections of posed 2D RGB images and 2D semantic maps, each collection describing an independent scene. Given a new set of posed 2D RGB images, we extract an implicit volumetric representation of the scene's 3D geometry and infer a 3D semantic field. The semantic field can then be used to render dense 2D semantic maps from novel camera poses or queried directly in 3D. Our method generalizes to novel scenes, and requires as little as one semantic map per scene at training time.

# 1 Introduction

High-level semantic understanding of 3D scenes as captured by digital images and videos is a fundamental objective of computer vision. Well-studied tasks such as scene classification (Krizhevsky et al., 2012), object detection (Liu et al., 2020a), semantic segmentation (Lateef & Ruichek, 2019), and instance segmentation (Hafiz & Bhat, 2020) infer semantic descriptions of scenes from RGB and other sensors and form the foundation for applications such as navigation (Bonin-Font et al., 2008) and robotic interaction (Billard & Kragic, 2019).

The most common approach to scene understanding is to narrow the scope to 2D (image-space) reasoning, wherein classical image-to-image architectures (Ronneberger et al., 2015) are trained on large collections of semantically-annotated images (Lin et al., 2014). These methods, while straightforward to apply, only produce per-pixel annotations and largely ignore the underlying 3D structure of the scene. Instead, our goal is to use a set of RGB images with known poses to produce a 3D *semantic field*: a function mapping 3D positions to probability distributions over semantic categories.

For 3D semantic segmentation, most prior work relies on 3D sensors (Izadi et al., 2011; Douillard et al., 2011; Kowdle et al., 2018) or 3D semantic labels (Dai et al., 2017; Caesar et al., 2020a). Convolutional network architectures have been designed for 3D point clouds (Qi et al., 2017a; Choy et al., 2019), voxel grids (Tran et al., 2015), and polygonal meshes (Hanocka et al., 2019; Masci et al., 2015). However, 3D sensors are not as affordable nor as widely available as RGB cameras, and 3D annotations are significantly more challenging to produce than their 2D counterparts and are hence generally scarce (Genova et al., 2021). Conversely, 2D sensors are ubiquitous. Therefore, the ability to reason about 3D information from 2D supervision alone is crucial to the deployment of 3D computer vision at scale.

To overcome the aforementioned challenges, researchers have adopted *hybrid* 2D/3D reasoning, so as to *propagate* densely supervised semantic signals from 2D projections to an underlying 3D substrate (Dai & Nießner, 2018; Genova et al., 2021; Kundu et al., 2020). At test time, these methods still require a classical 3D representation to be provided as input, hence limiting their applicability and performance. One interesting exception is Atlas (Murez et al., 2020), which only requires posed photos at test time but still requires 3D supervision to train.

In parallel to these developments, a new family of 3D representations have emerged based on coordinate-wise functions (Xie et al., 2021). In this regime, one trains a neural network to predict quantities such as occupancy,

signed distance, density and radiance. However, the vast majority of methods built on this approach target computer graphics applications such as novel view synthesis (Wang et al., 2021a; Zhang et al., 2021; Liu et al., 2020b; Rebain et al., 2020; Yu et al., 2021c) without an interpretable semantic understanding of the scene. One notable exception is Semantic-NeRF (Zhi et al., 2021), which regresses a per-3D point semantic class in addition to radiance and density. Semantic-NeRF produces novel *views* given a set of input views for a scene. Generalization to novel scenes is achieved by providing pseudo-labeled 2D semantic maps as input, which are fused by a NeRF-like model (Mildenhall et al., 2020) to produce labeled novel views. Generalization across scenes is thus a consequence of the 2D CNN that is used for producing pseudo labels. While this scales efficiently, it biases the model to the same pitfalls as 2D CNN methods where texture and color are more prevalent than shape information for segmenting objects (Geirhos et al., 2018).

In this work, we introduce *Neural Semantic Fields (NeSF)*, a method for semantic segmentation of 3D scenes via image-space semantic annotations; see Figure 1. In place of an explicit 3D representation, NeSF builds on the implicit representations of geometry recovered by methods such as NeRF. In particular, we apply a neural network to NeRF's *density field* to predict what we refer to as a scene's *semantic field*. A semantic field is thus defined as a coordinate-wise function mapping 3D points to probability distributions over semantic categories. Similar to Semantic-NeRF, the closest contemporary method, we apply the volumetric rendering equation to generate 2D semantic maps, enabling supervision from posed semantic annotations in image-space. Unlike Semantic-NeRF, NeSF's 3D segmentations are derived directly from a scene's inferred 3D geometry. To the best of our knowledge, NeSF is the first method capable of producing dense 2D and 3D segmentations of novel scenes from only posed RGB images.

As large scale datasets of 3D semantically annotated scenes with sufficient high quality RGB views are scarce, we propose three novel datasets of increasing complexity: `KLEVR`, `ToyBox5`, and `ToyBox13`. While datasets for 2D and 3D semantic scene understanding already exist (Dai et al., 2017; Chang et al., 2017; Straub et al., 2019; Caesar et al., 2020b), they presently lack the scale, detail, and precision necessary to to produce high-quality NeRF reconstructions. In particular, ScanNet suffers from issues with motion blur and exposure variation, MatterPort3D panoramas are too sparsely captured, and Replica contains too few scenes. Though early attempts to resolve issues such as image blur (Ma et al., 2021) have been made, we consider the additional method development required to successfully train NeRF representations on such scenes as beyond the scope of this work. We construct over 1,000 scenes of randomly-placed objects and render hundreds of RGB images with realistic lighting and materials. Notably, random object placement breaks relational consistencies between objects that exist in available datasets (e.g. chairs are likely to be found in the neighborhood of tables), adding an additional level of difficulty. Each RGB image is paired with corresponding ground-truth camera intrinsics and extrinsics, a semantic map and a depth map. We evaluate our method on these three datasets and compare its performance to competitive techniques in 2D and 3D scene understanding.

**Contributions**.

- We introduce the first method for generating 3D semantic fields from the geometry of neural fields, trained solely on posed RGB images and semantic maps. Unlike prior work, our method (i) can be queried anywhere within a bounded 3D volume, (ii) is capable of rendering semantic maps from novel camera poses, and (iii) generalizes to novel scenes with as few as one semantic map per scene at training time.
- We propose three new synthetic datasets for 2D and 3D semantic scene understanding. In total, these datasets are comprised of over 1,000 scenes and 3,000,000 realistically-rendered and semantically annotated frames. We release these datasets along with code to reproduce them to the community upon publication.

## 2 Related works

We now briefly overview related work in semantic segmentation (Minaee et al., 2021; He et al., 2021) and 3D reconstruction (Jin et al., 2020).

**Semantic segmentation**. Semantic segmentation is a heavily researched area, with most methods targeting a fully supervised, single modality problem (2D: Minaee et al. (2021) or 3D: He et al. (2021)). 2D approaches like DeepLab (Chen et al., 2017) train a CNN to segment each pixel in an image. There are also analogous

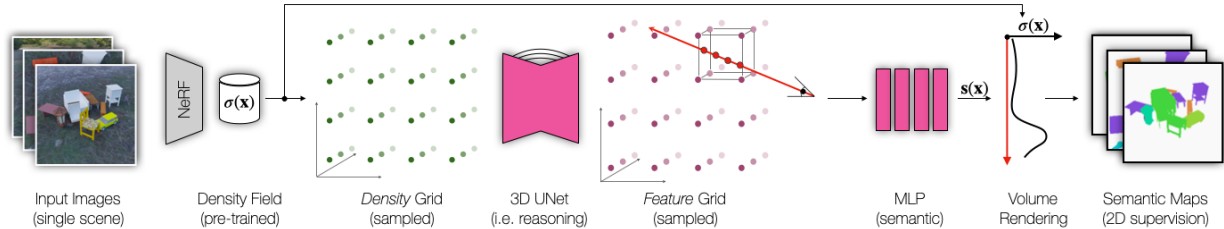

Figure 2: **Architecture** – Given a pre-trained NeRF model, we sample its volumetric density grid to obtain the 3D scene representation. This grid is converted to a semantic-feature grid by employing a fully convolutional volume-to-volume network thus allowing for geometric reasoning. The semantic-feature grid is in turn translated to semantic probability distributions using the volumetric rendering equation. Note the semantic 3D UNet is trained across all scenes in the TRAIN SCENES set, though not explicitly depicted for the sake of simplicity. Additionally, note that NeSF is trained *solely* using 2D supervisory signals and that no segmentation maps are provided at test time.

approaches in 3D for various shape representations – point clouds (Qi et al., 2017b; Hu et al., 2020), sparse or dense voxel grids (Dai & Nießner, 2018; Graham et al., 2018; Han et al., 2020), or meshes (Hanocka et al., 2019; Huang et al., 2019). In contrast to these methods, our method reconstructs and then segments a dense 3D representation from 2D inputs and supervision alone, and does not require ground truth 3D annotations or input geometry.

**Hybrid and multi-modal methods**. Many methods use one data modality to supervise or inform another (Ephrat et al., 2018; Koepke et al., 2020; Alwassel et al., 2020; Nagrani et al., 2020; Gupta et al., 2016; Jing et al., 2020). For 3D semantic segmentation, multiview fusion (Hermans et al., 2014; Kundu et al., 2020; Armeni et al., 2019; McCormac et al., 2017; Mascaro et al., 2021; Zhang et al., 2019) is a popular family of methods that requires only image supervision. However, these methods reason exclusively in the image domain and require an input 3D substrate such as a point cloud or polygonal mesh on which to aggregate 2D information. Similarly, Genova et al. (2021) propose a method for 3D point cloud segmentation from 2D supervision, but still require input 3D geometry. In an independent line of work, researchers have proposed pipelines for 3D segmentation that benefit from 2D image features (Kundu et al., 2020; Jaritz et al., 2019; Dai & Nießner, 2018; Vora et al., 2020). Unlike our method, these approaches also require full 3D supervision.

**Implicit representations**. Similar to our approach, Atlas (Murez et al., 2020) simultaneously learns a 3D implicit TSDF reconstruction and its scene segmentation from posed 2D images. However, this approach requires ground truth 3D data and supervision, while our method requires only images at *both* train and test time. Other methods use implicit representations to reconstruct a 3D scene (Sucar et al., 2021; Peng et al., 2020) or shape (Chen et al., 2020; Deng et al., 2020; Mescheder et al., 2019; Chen & Zhang, 2019; Niemeyer et al., 2020; Peng, 2021; Genova et al., 2019; Oechsle et al., 2021; Park et al., 2019; Takikawa, 2021). A more recent work approaches the problem with image supervision alone (Sucar et al., 2021) but does not consider semantics.

**Neural radiance fields**. Recently, a variety of methods based on NeRF (Mildenhall et al., 2020) have become popular for novel view synthesis (Martin-Brualla et al., 2021; Barron et al., 2021; Jeong, 2021; Lin et al., 2021; Wang et al., 2021a), 3D reconstruction (Rematas et al., 2021; Yariv et al., 2020; Jain et al., 2021; Jang & Agapito, 2021; Henzler, 2021; Yu et al., 2021c; Yuan et al., 2021; Chen & Xu, 2021), generative modeling (Meng, 2021; Schwarz et al., 2020; Niemeyer & Geiger, 2021) and semantic segmentation (Zhi et al., 2021). The majority of these models demonstrate impressive results but are only applicable in the single-scene setting. Other methods generalize to novel scenes but focus on novel view synthesis or reconstruction. Most relevant to our method is Semantic-NeRF (Zhi et al., 2021). Using ground-truth or pseudo-labeled semantic maps, it leverages the multi-view fusion capabilities of NeRF to generate 2D semantic predictions from novel views. In contrast to prior work, our approach directly generalizes to novel scenes using the 3D geometry extracted from the neural fields.

## 3    Method

We train NeSF on a collection of $S$ scenes, each described by a collection of RGB images $\{\mathcal{C}_{s,c}^{\text{gt}} \in [0,1]^{H \times W \times 3}\}$ and paired to a collection of semantic maps $\{\mathcal{S}_{s,c}^{\text{gt}} \in \mathbb{Z}_+^{H \times W}\}$. Both images and maps are indexed by camera index $c$ and a scene index $s$. For the sake of exposition, we assume that each RGB map is paired with a semantic map and that each scene contains $C$ map pairs, but the method itself makes no such assumption. Similar to prior work, we also assume the availability of camera calibration parameters $\{\gamma_{s,c} \in \mathbb{R}^\Gamma\}$ providing an explicit connection between each pixel and the 3D ray $\mathbf{r}$ cast within the 3D scene. We consider the problem of jointly estimating the camera calibration and the scene representation outside the context of this work, see recent approaches (Wang et al., 2021b; Lin et al., 2021; Yen-Chen et al., 2021). Our method involves two stages, which are described in the following subsections:

- **Section 3.1**: In the first stage, we pre-train neural radiance fields on posed RGB maps $\{(\mathcal{C}_{s,c}^{\text{gt}}, \gamma_{s,c})\}$ *independently* for each scene $s \in [1 \cdots S]$. This results in a set of neural radiance fields with network parameters $\{\theta_s\}$. To focus on the core task of understanding from 3D geometry, we disregard the radiance portion of these fields and employ the *volumetric density fields* $\boldsymbol{\sigma}(\mathbf{x} \mid \theta_s) \in [0, \infty)$ below.

- **Section 3.2**: In the second stage, we train a density-to-semantics *translation* network $\mathcal{T}$ parameterized by $\boldsymbol{\tau} = \{\boldsymbol{\tau}_{\text{unet}}, \boldsymbol{\tau}_{\text{mlp}}\}$. Given a scene's 3D geometry represented by density field $\boldsymbol{\sigma}_s = \boldsymbol{\sigma}(\cdot \mid \theta_s)$, this network produces a 3D semantic field $\mathbf{s}(\mathbf{x} \mid \boldsymbol{\sigma}_s, \boldsymbol{\tau})$ assigning each point a probability distribution over semantic categories. While the translation network produces a 3D field, we apply the volumetric rendering equation to obtain 2D semantic maps from reference camera poses $\{\gamma_{s,c}\}$. Predicted semantic maps can then be compared to their ground truth counterparts $\{\mathcal{S}_{s,c}^{\text{gt}}\}$ in a differentiable way.

### 3.1    NeRF pre-training

To extract an accurate, dense representation of each scene's 3D geometry, we leverage neural radiance fields as proposed in Mildenhall et al. (2020). To simplify notation, we drop scene index $s$ for the remainder of this section as all scenes can be trained independently and in parallel. More specifically, given a collection of posed RGB images $\{(\mathcal{C}_c^{\text{gt}}, \gamma_c)\}$, and denoting with $\mathbf{r} \sim \mathcal{R}(\gamma_c)$ rays corresponding to pixels from image $\mathcal{C}_c^{\text{gt}}$, a neural radiance field model with parameters $\theta$ is trained by minimizing the squared photometric reconstruction loss:

$$\mathcal{L}_{\text{rgb}}(\boldsymbol{\theta}) = \sum_c \mathbb{E}_{\mathbf{r} \sim \mathcal{R}(\gamma_c)} \left[ ||\mathcal{C}(\mathbf{r} \mid \theta) - \mathcal{C}_c^{\text{gt}}(\mathbf{r})||_2^2 \right] \tag{1}$$

where $\mathcal{C}_c^{\text{gt}}(\mathbf{r})$ is the ground truth color of ray passing through a pixel in image $c$, and the color $\mathcal{C}(\mathbf{r} \mid \theta)$ is computed by applying the volumetric rendering equation with the ray's near and far bounds $t \in [t_n, t_f]$:

$$\mathcal{C}(\mathbf{r} \mid \theta) = \int_{t_n}^{t_f} w(t \mid \theta) \cdot \mathbf{c}(t \mid \theta) \, dt \tag{2}$$

Let $\mathbf{r}(t) = \mathbf{o} + t\mathbf{d}$ represent a point along a ray with origin $\mathbf{o}$ and direction $\mathbf{d}$. Weight $w(t) = w(\mathbf{r}(t))$ is then defined as:

$$w(t) = \underbrace{\exp\left(-\int_{t_n}^{t} \boldsymbol{\sigma}(u) \, du\right)}_{\text{visibility of } \mathbf{r}(t) \text{ from } \mathbf{o}} \cdot \underbrace{\boldsymbol{\sigma}(t)}_{\text{density at } \mathbf{r}(t)} \tag{3}$$

where the volumetric density $\boldsymbol{\sigma}(t)$ and radiance fields $\mathbf{c}(t)$ are predicted by a multi layer perceptron (i.e. MLP) with Fourier feature encoding. We refer the reader to the original work (Mildenhall et al., 2020) for further details and the discretization of these integrals (Max, 1995).

**Training**. While neural radiance fields are acknowledged to be slow to train, we find that we are able to fit a single model to sufficient quality in $\approx$20 minutes on eight TPUv3 cores on the Google Cloud Platform. Once trained, the per scene parameters $\{\theta_s\}$ are held fixed. While we acknowledge the slow training speed of neural radiance fields, we utilize it as an initial proof-of-concept for the proposed method. Within the research community, several efforts are targeting efficiently speeding up and scaling NeRF-like models (Müller et al., 2022; Yu et al., 2021a). We view extending NeSF to utilize information generated by such methods as future work.

### 3.2 Semantic Reasoning

We now present a method for mapping 3D density fields to 3D semantic fields. To recap, we train a *translation model* $\mathcal{T}(\boldsymbol{\sigma} \,|\, \boldsymbol{\tau})$ to produce a semantic field $\mathbf{s}(\mathbf{x} \,|\, \boldsymbol{\sigma}, \boldsymbol{\tau})$ being given access to the density field of a scene $\boldsymbol{\sigma}$, where $\mathbf{s}$ assigns a probability distribution over semantic categories at each 3D point in space. We optimize translation model's parameters $\boldsymbol{\tau}$ with 2D annotations alone. Our inspiration is drawn from methods that translate *explicit* representations of geometry into semantics (Qi et al., 2017a; Dai et al., 2018), and in observing that density fields provide an *implicit* notion of geometry. For ease of exposition, the overall architecture illustrated in Figure 2, is broken down into several discrete steps: ① density grid extraction, ② spatial reasoning, ③ feature decoding, ④ supervision, and ⑤ data augmentation.

**Density grid extraction**. Our method starts by uniformly evaluating the density field on a 3D lattice with spacing $\epsilon$ between samples:

$$\Sigma_s = \Sigma(\boldsymbol{\sigma}_s) = \{\boldsymbol{\sigma}(\mathbf{x} \,|\, \theta_s) \text{ s.t. } \mathbf{x} \in [-1 : \epsilon : +1]^3\} \tag{4}$$

While this operation limits the spatial resolution of the original density field, it presents a natural representation for further processing.

**Spatial reasoning (3D)**. We apply a 3D UNet (Çiçek et al., 2016) to $\Sigma_s$ to obtain a *feature* grid $\mathcal{F}_s$ of the same spatial resolution as $\Sigma_s$:

$$\mathcal{F}_s = \mathcal{F}(\Sigma_s | \boldsymbol{\tau}_{\text{unet}}) = \text{UNet3D}(\Sigma_s \,|\, \boldsymbol{\tau}_{\text{unet}}) \tag{5}$$

This step is essential as a point-wise measurement of the density field $\boldsymbol{\sigma}(\mathbf{x})$ does not contain sufficient information to capture 3D structure. After all, $\boldsymbol{\sigma}(\mathbf{x})$ only measures the volumetric density at a point, while the 3D structure requires reasoning over local spatial neighborhoods. Note that we *share* translation network parameters $\boldsymbol{\tau}_{\text{unet}}$ across scenes, thereby enabling the network to learn features *across* scenes as well as facilitating generalization to *novel* scenes without needing pre-labeled images in input.

**Feature decoding**. Given a query point $\mathbf{x} \in \mathbb{R}^3$, we interpolate within the feature grid $\mathcal{F}_s$ to obtain a feature vector corresponding to $\mathbf{x}$. We then employ a neural network decoder $\mathcal{D}$ to generate a field of probability distributions over semantic categories:

$$\mathbf{s}(\mathbf{x} \,|\, \mathcal{F}_s, \boldsymbol{\tau}_{\text{mlp}}) = \mathcal{D}(\text{TriLerp}(\mathbf{x}, \mathcal{F}_s) \,|\, \boldsymbol{\tau}_{\text{mlp}}) \tag{6}$$

where $\mathcal{D}$ is a multilayer perceptron with trainable parameters $\boldsymbol{\tau}_{\text{mlp}}$ and TriLerp applies trilinear interpolation similar to (Liu et al., 2020b). Like their UNet counterpart, parameters $\boldsymbol{\tau}_{\text{mlp}}$ are *shared* across all scenes.

**Supervision**. To supervise the training of parameters $\boldsymbol{\tau}$, we employ volumetric rendering as in NeRF (Mildenhall et al., 2020) but adapt it to render semantic maps as in Semantic-NeRF (Zhi et al., 2021):

$$\mathcal{S}(\mathbf{r} \,|\, \boldsymbol{\sigma}_s, \boldsymbol{\tau}) = \int_{t_n}^{t_f} w(t \,|\, \boldsymbol{\sigma}_s) \cdot \mathbf{s}(\mathbf{x} \,|\, \boldsymbol{\sigma}_s, \boldsymbol{\tau}) \, dt \tag{7}$$

We supervise the training process for $\boldsymbol{\tau}$ by minimizing the softmax cross-entropy between rendered semantic and ground truth semantic maps:

$$\mathcal{L}_{\text{sem}}(\boldsymbol{\tau}) = \mathbb{E}_s \left[ \mathcal{L}_{\text{sem}}(\theta_s, \boldsymbol{\tau}) \right] \quad \text{where:} \tag{8}$$

$$\mathcal{L}_{\text{sem}}(\theta_s, \boldsymbol{\tau}) = \sum_c \mathbb{E}_{\mathbf{r} \sim \mathcal{R}(\gamma_c)} \left[ \text{CE}(\mathcal{S}(\mathbf{r} \,|\, \theta_s, \boldsymbol{\tau}), \mathcal{S}_c^{\text{gt}}(\mathbf{r})) \right]$$

We include an additional smoothness regularization term to encourage similar predictions in local neighborhoods. We sample points $\mathbf{x} \sim \text{Uniform}([-1, 1]^3)$ and normally distributed noise $\epsilon \sim \mathcal{N}(0, 0.01)$,

$$\mathcal{L}_{\text{reg}}(\boldsymbol{\tau}) = \mathbb{E}_{\mathbf{x}, \epsilon} \left[ ||\mathbf{s}(x | \mathcal{F}, \boldsymbol{\tau}) - \mathbf{s}(x + \epsilon | \mathcal{F}, \boldsymbol{\tau})||_2^2 \right] \tag{9}$$

Our total loss is thus $\mathcal{L}(\boldsymbol{\tau}) = \mathcal{L}_{\text{sem}}(\boldsymbol{\tau}) + \lambda \mathcal{L}_{\text{reg}}(\boldsymbol{\tau})$.

Table 1: **Dataset statistics** – Each dataset consists of a set of train and novel scenes, wherein each scene's cameras are split into a train and test set (denoted by a "/").

| | KLEVR | ToyBox5 | ToyBox13 |
|---|---|---|---|
| # scenes | 100 / 20 | 500 / 25 | 500 / 25 |
| # cameras/scene | 210 / 90 | 210 / 90 | 210 / 90 |
| # total cameras | 36,000 | 1,575,00 | 1,575,00 |
| frame resolution | $256 \times 256$ | $256 \times 256$ | $256 \times 256$ |
| # objects/scene | 4-12 | 4-12 | 4-12 |
| # object instances | 5 | 25,905 | 39,695 |
| # background instances | 1 | 383 | 383 |

**Data augmentation**. To increase the robustness of our method and similarly to classical methods (Qi et al., 2017a;b), we apply data augmentation in the form of random rotations around the z-axis (i.e. upwards). In particular, we randomly sample an angle $\gamma \in [0, 2\pi)$ at each step of training. Rather than extracting a density grid in NeRF's original coordinate system, we construct a rotation transformation $R_\gamma$ and query NeRF at points $\mathbf{x}' = R_\gamma(\mathbf{x})$, resulting in the following density grid:

$$\tilde{\Sigma}_s = \{\boldsymbol{\sigma}(\mathbf{x}' | \theta_s) \text{ s.t. } \mathbf{x} \in R_\gamma^{-1}([-1 : \epsilon : +1]^3))\} \tag{10}$$

Subsequently, the spatial reasoning is applied to the rotated density field to produce a feature grid. To query the feature grid, the ray of interest is similarly rotated in order to sample semantic features appropriately. The remainder of the algorithm remains as before. Note that this procedure does not necessitate the retraining of NeRF models.

## 4   Datasets – Table 1 and Figure 3

To investigate NeSF, we require datasets describing the appearance and semantics of a large number of scenes from multiple points of view. While existing datasets based on indoor and self-driving sensor captures exist (Dai et al., 2017; Chang et al., 2017; Caesar et al., 2020b; Sun et al., 2020), we desire a controlled setting where distractors such as motion blur, camera calibration error, and object motion can be eliminated.     To this end, we introduce three new datasets built on Kubric (Greff et al., 2022): KLEVR, ToyBox5, and ToyBox13. Each dataset consists of hundreds of synthetic scenes, each containing randomly-placed 3D objects which are photo-realistically rendered by a path tracer (Blender, 2018) supporting soft shadows, physically based materials, and global illumination effects. Each scene is described by a set of posed frames, where each frame provides an RGB image, a semantic map, and a depth map rendered from a shared camera pose. We provide the Kubric script to generate such scenes, so as to enable follow-up research to build increasingly challenging datasets. Figure 3 depicts images from sampled scenes of the generated datasets.

KLEVR. We design the KLEVR dataset to be a simple testbed akin to MNIST in machine learning. Inspired by CLEVR (Johnson et al., 2017), each scene contains 4 to 12 simple geometric objects placed randomly on a matte grey floor. Each object is assigned a random hue and scale, and is constrained to lie within a fixed bounding box. The semantic category of an object is set equal to its geometry class (e.g. cube, cylinder, etc). While only the shape of each object is semantically relevant, color, scale, and placement serve as distractors. For each scene, we render 300 frames from randomly-sampled camera poses aimed at the scene's origin. Camera poses are constrained to lie in the upper hemisphere surrounding the scene. For each frame, we render an RGB image, a semantic map, and a depth map; the latter is solely used by some of the compared baselines, and for computing evaluation metrics.

ToyBox5 and ToyBox13. These datasets are designed to imitate scenes of children's bedrooms and are designed to be more challenging. Scenes are constructed from a large vocabulary of ShapeNet (Chang et al., 2015) objects coupled with HDRI backdrops (floor, horizon, and environment illumination) captured in the wild (Zaal et al., 2021). Like KLEVR, each scene consists of 4-12 randomly-placed objects, and frames are rendered from 300 independently-sampled inward-facing camera poses. Objects are sampled at random from the 5 and 13 most common object categories, respectively for ToyBox5 and ToyBox13. Such splits have been commonly used in the 3D deep learning literature (Groueix et al., 2018; Mescheder et al., 2019; Genova et al., 2019; Deng et al., 2020). Like the objects themselves, backdrops are sampled at random when constructing a

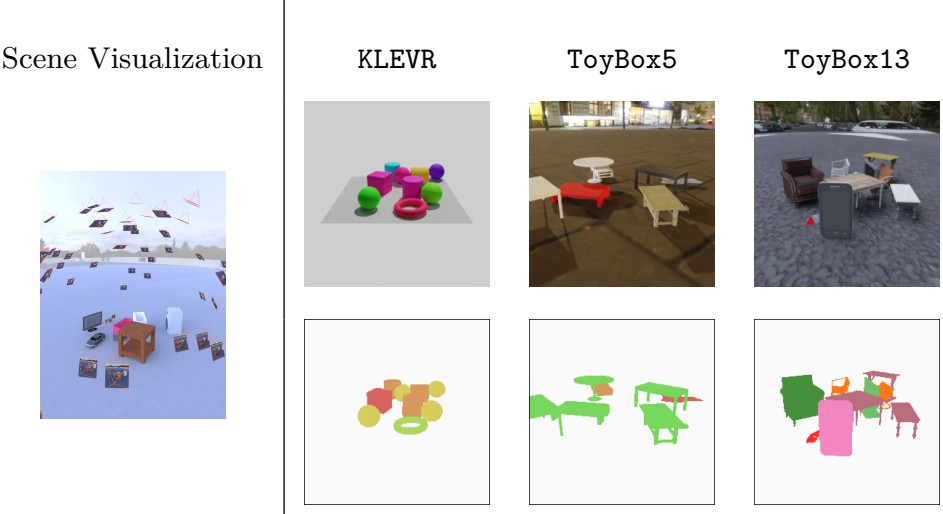

Figure 3: **Dataset examples** – (Left) The frames are rendered from independently-sampled inward-facing camera poses. (Right) Each frame includes an RGB image, semantic map, and depth map (not pictured here).

scene. With thousands of objects per category to choose from, most object instances appear rarely or only once.

**Train/test splits**. To enable evaluation from novel views within the same scene, we randomly partition each scene's frames into TRAIN CAMERAS and TEST CAMERAS; the latter representing the set typically used to evaluate methods in novel view synthesis (Mildenhall et al., 2020). For evaluation *across* scenes, we further partition scenes into TRAIN SCENES and NOVEL SCENES. Table 1 depicts the statistics for each of the proposed datasets.

## 5 Experiments

### 5.1 Setup

We evaluate NeSF on the three datasets described in Section 4. Unless otherwise stated, we train NeRF models on all TRAIN CAMERAS from all TRAIN SCENES. We provide NeSF with supervision from semantic maps corresponding to 9 randomly-chosen cameras per scene. These frames comprise approximately 4% of all RGB frames and strike a balance between multi-view information and a label-scarce regime. For 2D evaluation, we randomly select 4 cameras from each NOVEL SCENES' TRAIN CAMERAS. For 3D evaluation, we use camera parameters and ground truth depth maps to derive a labeled 3D point cloud from the same 4 cameras. Semantic segmentations are evaluated according to 2D and 3D mean intersection-over-union. Further details regarding the training setup are provided in the supplementary material.

**Training Details**. Each scene is preprocessed by training an independent NeRF for $25k$ steps with Adam using an initial learning rate of $1e-3$ decaying to $5.4e-4$ according to a cosine rule. Our NeRF architecture follows the original work. NeSF is trained for $5k$ steps using Adam with an initial learning rate of $1e-3$ decaying to $4e-4$. As input for NeSF, we discretize density fields by densely probing with $\epsilon=1/32$ resulting in $64^3$ evenly-spaced points in $[-1, +1]^3$. This density grid is then processed by the 3D UNet architecture of Çiçek et al. (2016) with 32, 64, and 128 channels at each stage of downsampling. The semantic latent vector is processed by a multilayer perceptron consisting of 2 hidden layers of 128 units. Our models are trained on 32 TPUv3 cores. We intend to release the code, datasets and pre-trained models upon publication.

**Segmentation baselines (2D/3D)**. We compare NeSF to two popular semantic segmentation baselines, DeepLab (Chen et al., 2017) and SparseConvNet (Graham et al., 2018). The aforementioned methods were

| | TRAIN CAMERAS | | TEST CAMERAS | |
| | 2D mIoU | 3D mIoU | 2D mIoU | 3D mIoU |
|---|---|---|---|---|
| NeSF | $92.7 \pm 0.2$ | $\mathbf{97.8 \pm 0.7}$ | $92.6 \pm 0.2$ | $\mathbf{97.5 \pm 0.7}$ |
| Semantic-NeRF(DeepLab) | – | – | $\mathbf{93.7 \pm 0.002}$ | $90.0 \pm 0.007$ |
| DeepLab | $\mathbf{97.1}$ | – | – | – |
| Semantic-NeRF(Ground Truth) | – | – | $95.3 \pm 0.002$ | $93.2 \pm 0.009$ |
| SparseConvNet(Ground Truth Depth Maps) | – | $99.7$ | – | $99.7$ |

Table 2: **Quantitative comparison (KLEVR)** – NeSF is competitive with 2D and 3D baselines. Configurations marked as "–" denotes a setting where methods are not applicable. As DeepLab can only be queried given an RGB input, it is not evaluated on the TEST CAMERAS. Semantic-NeRF, unlike NeSF, is trained on a per-scene basis, thus we report the numbers on the TEST CAMERAS only. The double line separating methods is to highlight approaches that utilize a different data modality compared to NeSF. Semantic-NeRF trained on ground-truth 2D semantics has access to ground-truth semantic labels for all TRAIN CAMERAS. SparseConvNet utilizes 3D supervision in the form of a 3D labeled point cloud. Note that all methods here, with the exception of Semantic-NeRF, are trained using semantic supervision generated from only 9 images per-scene.

selected due to their competitive performance, and their ease of application. DeepLab follows a traditional 2D semantic segmentation pipeline, producing segmentation maps from RGB images. We train DeepLab v3 with Wide ResNet (Wu et al., 2019) for $55k$ steps on 16 TPUv3 chips. SparseConvNet is a point cloud segmentation method and, unlike NeSF and DeepLab, requires explicit 3D supervision. We train SparseConvNet asynchronously on 20 NVIDIA V100 GPUs with momentum using a base learning rate of $1.5e{-2}$ and decaying to 0 over the final $250k$ steps of training. We refer the reader to Section 5.2 and the supplementary material for further details.

**Semantic-NeRF**. We compare NeSF to Semantic-NeRF (Zhi et al., 2021), which simultaneously optimizes a NeRF (Mildenhall et al., 2020) reconstruction while also fusing 2D semantic maps from the same views. While Semantic-NeRF is evaluated on 2D mIoU of unseen views, we compare with NeSF by evaluating both 2D mIoU as well as 3D mIoU of novel scenes. We compare Semantic-NeRF's performance with NeSF in two distinct regimes: 1) using pseudo-labeled semantic maps generated by a 2D segmentation method, and 2) using ground-truth semantic maps. While the former are easy to acquire, pseudo-labeled semantic maps may contain multi-view inconsistencies and other errors. On the other hand, it is infeasible to obtain ground-truth labels for all test scene images in a practical application, and this regime serves as an upper bound on Semantic-NeRF's performance.

Note that in each of these scenarios, one method is at a disadvantage compared to the other. Training Semantic-NeRF using ground-truth semantic labels puts NeSF at a disadvantage in terms of label scarcity, as NeSF does not use ground-truth labels for NOVEL SCENES and only uses a few labeled semantic maps for TRAIN SCENES. At the same time, training Semantic-NeRF using pseudo labeled semantic maps puts NeSF at an advantage as pseudo labeled semantic maps can contain mislabeled regions which in turn affect the final method's performance. Nonetheless, as Semantic-NeRF shares a number of commonalities with NeSF, we report the numbers in both regimes with the goal of presenting and understanding the limitations of each method.

## 5.2 Comparisons to baselines

Our first set of experiments evaluates the performance of our proposed method in comparison to baselines on the KLEVR, ToyBox-5, and ToyBox-13 datasets.

**DeepLab (Chen et al., 2017) (2D)**. To maintain a fair comparison in 2D, we train both the semantic phase of NeSF and DeepLab on an identical set of paired RGB images and semantic maps for a fixed set of scenes (i.e. 9 per scene in TRAIN SCENES). NeSF has further access to all 210 RGB maps associated with each scene's TRAIN CAMERAS, which are used to fit per-scene NeRF models. Both methods are evaluated on a random sampling of frames from NOVEL SCENES (i.e. 4 per scene). To emphasize the 3D nature of

| | ToyBox5 | | | | ToyBox13 | | | |
|---|---|---|---|---|---|---|---|---|
| | TRAIN CAMERAS | | TEST CAMERAS | | TRAIN CAMERAS | | TEST CAMERAS | |
| | 2D mIoU | 3D mIoU | 2D mIoU | 3D mIoU | 2D mIoU | 3D mIoU | 2D mIoU | 3D mIoU |
| NeSF | 81.9 ± 0.8 | **88.7 ± 0.9** | **81.7 ± 0.6** | **89.6 ± 0.7** | 56.5 ± 0.8 | **60.1 ± 0.6** | **56.6 ± 1.0** | **61.9 ± 0.9** |
| Semantic-NeRF(DeepLab) | – | – | 72.3 ± 0.1 | 80.8 ± 0.1 | – | – | 55.1 ± 0.1 | 58.2 ± 0.2 |
| DeepLab | **81.6** | – | – | – | **63.1** | – | – | – |
| Semantic-NeRF(Ground Truth) | – | – | 84.2 ± 0.1 | 93.4 ± 0.1 | – | – | 81.1 ± 0.1 | 91.6 ± 0.1 |
| SparseConvNet(Ground Truth Depth Maps) | – | 93.4 | – | 94.8 | – | 83.2 | – | 81.7 |

Table 3: **Quantitative comparison (`ToyBox5`, `ToyBox13`)** – NeSF is competitive with 2D and 3D baselines. At train time, NeSF, Semantic-NeRF and DeepLab only utilize 2D supervision. Conversely, SparseConvNet requires full 3D supervision in the form of labeled 3D point clouds. We construct oracle point clouds via back-projected depth maps, resulting in an upper bound to our method. Models are evaluated on train and test camera poses from test scenes. Configurations marked as "–" denotes a setting where methods are not applicable. Statistics for NeSF and Semantic-NeRF are aggregated across five random initializations. Note that all methods here, with the exception of Semantic-NeRF, are trained using semantic supervision generated from only 9 images per-scene.

| | | Fixed-Color | Mono-Color | Color-Color | Texture |
|---|---|---|---|---|---|
| NeSF | 2D | **93.7 ± 2.0** | **92.3 ± 6.0** | **93.4 ± 2.0** | **92.5 ± 3.0** |
| | 3D | **86.0 ± 3.0** | **85.6 ± 2.0** | **88.3 ± 3.0** | **90.5 ± 5.4** |
| Semantic-NeRF (DeepLab) | 2D | 22.7 ± 0.0 | 37.7 ± 0.01 | 89.6 ± 0.0 | 40.5 ± 0.0 |
| | 3D | 22.0 ± 0.0 | 35.2 ± 0.0 | 85.6 ± 0.1 | 41.7 ± 0.0 |
| Semantic-NeRF (Ground Truth) | 2D | 95.0 ± 0.0 | 95.0 ± 0.0 | 94.6 ± 0.0 | 95.1 ± 0.0 |
| | 3D | 93.7 ± 0.0 | 92.5 ± 0.0 | 94.8 ± 0.0 | 94.4 ± 0.0 |

Table 4: **Quantitative comparison on `KLEVR` dataset perturbations** – Unlike Semantic-NeRF, NeSF is robust to appearance and color variations in the data as it relies on *geometry* for reasoning about semantics.

NeSF, we evaluate on additional camera poses from NOVEL SCENES where RGB information is not available. More precisely, for every scene, we evaluate on an additional 4 TEST CAMERAS. Note that this setting is not directly comparable to DeepLab as it requires an RGB input to produce the semantic predictions.

**SparseConvNet (Graham et al., 2018) (3D)**. As SparseConvNet requires 3D input, we derive an oracle point cloud for each scene from camera poses and *ground truth* depth maps – hence giving an *unfair advantage* to this method, establishing an upper bound on performance given full 3D supervision. For this, we use the same 210 train frames used to fit NeRF models in the 2D comparison. We further select a subset of each point cloud for 3D semantic supervision; namely, the points corresponding to the 9 semantic maps supervising NeSF and DeepLab. We evaluate NeSF and SparseConvNet on two sets of 3D points on NOVEL SCENES. The first set is a subset of each point cloud corresponding to 4 randomly chosen frames from each scene's TRAIN CAMERAS. These points are available to SparseConvNet as part of each scene's 3D representation. The second set is a set of additional query points derived from 4 additional frames from each scene's TEST CAMERAS. As SparseConvNet is not designed to classify points beyond its input point cloud, we apply a nearest neighbor label propagation procedure to assign labels to the latter.

**Semantic-NeRF (Zhi et al., 2021) (2D/3D)**. In order to present a complete picture of the performance of NeSF in comparison to Semantic-NeRF, we report numbers on two variations of Semantic-NeRF: 1) DeepLab: we use supervise Semantic-NeRF with semantic maps generated by DeepLab (Chen et al., 2017), and 2) Ground-truth: we use ground-truth 2D semantic maps. The DeepLab model used for pseudo-labeling is trained on the same set of semantic maps provided for training the semantic phase of NeSF. Specifically, the train dataset consists of 9 semantic maps per scene from each scene in TRAIN SCENES. For Semantic-NeRF fitting on NOVEL SCENES, either the DeepLab model is used to pseudolabel all 210 RGB maps associated with each scene's TRAIN CAMERAS, or ground-truth labels are supplied. Note that Semantic-NeRF is the only method that is trained using semantic supervision on all 210 TRAIN CAMERAS per-scene. Evaluation is carried out by evaluating on views for 4 unseen TEST CAMERAS for each of the NOVEL SCENES. Note that we do not report results for Semantic-NeRF on the TRAIN CAMERAS, as these views are used to train Semantic-NeRF

for each of NOVEL SCENES. In comparison, NeSF's semantic module *does not* require semantic supervision at test time.

**Quantitative comparisons** − **Table 2 and Table 3**. While all methods perform comparably on the KLEVR dataset, model quality varies drastically on the more challenging datasets. On ToyBox5, our method performs comparably to DeepLab, but on ToyBox13, it underperforms by 6.6% in 2D mIoU. While our method does not achieve the same level of accuracy as DeepLab on frames where RGB images are available, it is able to achieve near identical accuracy on *novel* camera poses, a task to which DeepLab cannot be applied. Compared to Semantic-NeRF trained with pseudo-labels on ToyBox5, our method outperforms by 9.4% and 8.8% in 2D and 3D mIoU respectively. We perform comparably on ToyBox13, with a slight improvement of 1.5% and 3.7% in 2D and 3D mIoU respectively. As expected, when comparing to Semantic-NeRF trained on ground-truth labels, we notice a larger performance gap, whereby NeSF underperforms by 2.5-3.8% on ToyBox5 and 24.5-29.7% on ToyBox13. As expected, our method also underperforms SparseConvNet by 4.7-5.2% on ToyBox5 and 19.8-23.1% on ToyBox13. Unlike SparseConvNet, our method lacks access to dense, ground truth depth maps and full 3D supervision. Further, the 3D UNet architecture employed by NeSF is based on (Çiçek et al., 2016), a predecessor to the SparseConvNet architecture. As NeSF does not take advantage of sparsity, it must operate at a lower spatial resolution than the baseline and tends to mislabel small objects and thin structures. Though NeSF underperforms SparseConvNet today, we anticipate methodological improvements in model architecture to rapidly improve performance. Additional in-depth analysis is included in the supplementary material.

**Qualitative comparisons** − **Figure 5 and Figure 6**. Qualitatively, our method exhibits strong performance in identifying the 13 canonical categories in ToyBox13. Because our method operates directly on 3D geometry, it is not easily confused by objects of similar appearance but dissimilar geometry as demonstrated by the thin standing rifle in the top row of Figure 5. While NeSF and SparseConvNet correctly recognize the rifle's geometry, DeepLab labels it identically to the chair behind it. Similar to DeepLab, our method faces challenges with thin structures such as the tube of the standing lamp pictured in the middle row. Such structures are not well captured by the regular grids employed by DeepLab and NeSF in 2D and 3D, respectively. With access to dense, accurate point clouds, SparseConvNet correctly identifies the lamp in its entirety. One limitation particularly evident in NeSF is a tendency to smear labels across nearby objects, as demonstrated by the chair partially labeled as a display in the bottom row. Without access to *appearance* or fine-grained geometry, NeSF is unable to identify when one object ends and another begins. Integrating appearance information and access to higher spatial resolution are potential ways to improve NeSF's accuracy.

In Figure 6, we compare the performance of NeSF and Semantic-NeRF (supervised by DeepLab pseudo-labels) on all datasets. Both NeSF and Semantic-NeRF can be seen to perform well on the least challenging KLEVR dataset, as the objects can easily be segmented in either 2D or 3D. However, on the more challenging ToyBox5 and ToyBox13 datasets, we find that while Semantic-NeRF captures some fine structures, the density fields generated by Semantic-NeRF contains a large amount of floaters which reduce the semantic accuracy of the model's predictions. Recently, methods such as Mip-NeRF-360 (Barron et al., 2022) propose regularization terms to reduce the presence of such artifacts. Floater artifacts are best visualized in rendered videos. To view side by side comparisons of such renders, we encourage the reader to view additional results in the accompanying video from timestamp 8:07. Furthermore, on both ToyBox5 and ToyBox13, we find that NeSF is more capable of capturing the fine details of the objects in the scene; like the structure of the lamp in ToyBox13 and the chair in ToyBox5. Similar to NeSF, Semantic-NeRF struggles to separate objects leading to some smearing artifacts in the semantic maps.

## 5.3 Ablation Studies

Our second set of experiments investigates how each component of our method affects system performance on the KLEVR dataset.

**Sensitivity to features**. Table 5 shows results of an ablation study where one feature of our method is varied while holding all others to their reference values. We find that each component provides a measurable improvement in 2D and 3D segmentation quality. Data augmentation in the form of random scene rotations

Table 5: **Ablation: NeSF hyperparameters** – Data augmentation, in the form of random scene rotations, increased spatial resolution of the density grid, and increased UNet model capacity improve 2D and 3D mIoU. Experiments on 25 scenes from the KLEVR dataset. Hyperparameters correspond to the following. *Random Rotations*: whether random rotations data augmention is applied to density grid during training, *Density Grid*: resolution of the density grid along the 3 spatial dimensions as referred to by epsilon in Equation 4, *UNet*: number of output features for each of 3 respective CNN layers of the UNet, and *MLP*: respective number of (hidden layers, hidden units) per layer of the MLP.

| Hyperparameter | | 2D | 3D |
|---|---|---|---|
| Random Rotations | No | 81.1 | 75.5 |
| | Yes | 92.0 | 97.1 |
| Density Grid | (32, 32, 32) | 87.5 | 92.1 |
| | (48, 48, 48) | 91.2 | 96.0 |
| | (64, 64, 64) | 91.7 | 89.6 |
| | (80, 80, 80) | 92.0 | 97.1 |
| UNet | (16, 32, 64) | 89.9 | 94.4 |
| | (24, 48, 96) | 91.5 | 96.4 |
| | (32, 64, 128) | 92.0 | 97.1 |
| MLP | (0, 32) | 91.3 | 96.4 |
| | (1, 32) | 91.8 | 96.9 |
| | (1, 64) | 91.2 | 96.2 |
| | (2, 128) | 92.0 | 97.1 |

Table 6: **Ablation: sensitivity to reconstruction quality** – The accuracy of our method improves with NeRF's reconstruction quality. PSNR and SSIM are are averaged across all scenes and metrics aggregated. Experiments on all scenes from KLEVR.

| # RGB | NeRF | | NeSF | |
|---|---|---|---|---|
| Images | PSNR | SSIM | 2D | 3D |
| 5 | $21.2 \pm 1.4$ | $0.89 \pm 0.02$ | 52.2 | 72.4 |
| 10 | $24.2 \pm 1.2$ | $0.92 \pm 0.01$ | 79.6 | 96.7 |
| 25 | $30.3 \pm 1.2$ | $0.96 \pm 0.01$ | 87.3 | 97.0 |
| 50 | $35.5 \pm 0.9$ | $0.98 \pm 0.00$ | 90.8 | 97.3 |
| 75 | $37.5 \pm 1.0$ | $0.98 \pm 0.00$ | 91.4 | 97.1 |
| 100 | $38.4 \pm 1.1$ | $0.98 \pm 0.00$ | 92.0 | 97.4 |

improves quality the most, adding 10.3% and 11.8% to 2D and 3D mIoU respectively. The spatial resolution of the probed NeRF density grids is the second most important as insufficient resolution makes smaller objects indistinguishable. We refer the reader to Figure 10 in the supplementary material for visualizations of the density grid.

**Sensitivity to reconstruction quality** – **Table 6**. We investigate the robustness of NeSF to NeRF reconstruction quality. To modulate reconstruction quality, we vary the number of RGB images used when fitting NeRF models from 5 to 100. As expected, NeRF reconstruction quality as measured on novel views increases as more RGB images are provided. At the same time, we find that NeSF 2D segmentation accuracy improves monotonically with NeRF reconstruction quality. Improvement of NeRF reconstruction quality is a highly active area of research, as such we anticipate methodological improvements may be directly applied to improve NeSF's performance. Surprisingly, 3D segmentation accuracy levels off near 97% when NeRF models are optimized with as little as 25 RGB images.

**Sensitivity to data scarcity** – **Figure 4**  To investigate NeSF's applicability to scenarios where labeled semantic maps are scarce, we investigate robustness to the number of semantic maps per scene. We find NeSF easily generalizes to novel scenes with *as few as one semantic map per train scene*. Additional semantic maps per scene improve performance given a small number of scenes, with no noticeable effect after 25 scenes. This suggests datasets consisting of videos, each with a single labeled frame, are ideal for NeSF.

We note, however, that performance does not monotonically increase with the number of semantic maps per scene, particularly in the regime of limited training scenes. While additional maps per scene will generally increase accuracy on held-out views of the training scenes, it is not guaranteed to improve performance on *novel scenes*.

**Robustness to dataset shift** – **Figure 8 and Table 4**

In the following, we design a set of experiments highlighting the differences between NeSF and Semantic-NeRF in semantic prediction of novel scenes. We hypothesize that NeSF uses orthogonal information to that used by Semantic-NeRF for learning the semantics of a scene. Specifically, NeSF learns semantics from 3D geometry allowing the method to be robust to visual, texture based changes commonly observed amongst every day objects (Geirhos et al., 2018). A model that possesses such a property is expected to perform more robustly

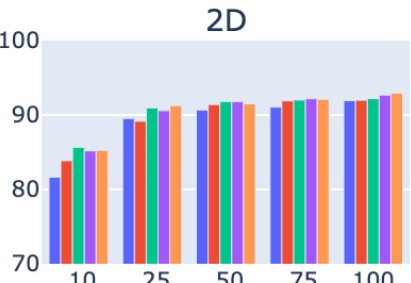 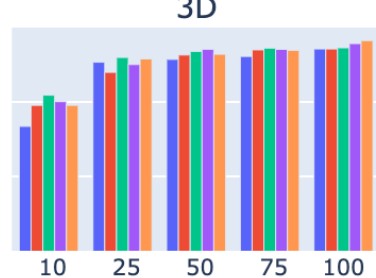

Figure 4: **Ablation: data efficiency** – 2D and 3D mIoU as a function of the number of train scenes for scenes with supervision from **1**, **2**, **5**, **10**, or **25** semantic maps per scene. NeSF generalizes to new scenes with as few as a *one semantic map per scene*. Additional semantic maps per scene marginally improve the accuracy. Experiments on `KLEVR` dataset. While performance on novel scenes is not monotonic with respect to the number of semantic maps per scene, it is nearly monotonic with respect to number of training scenes, as expected.

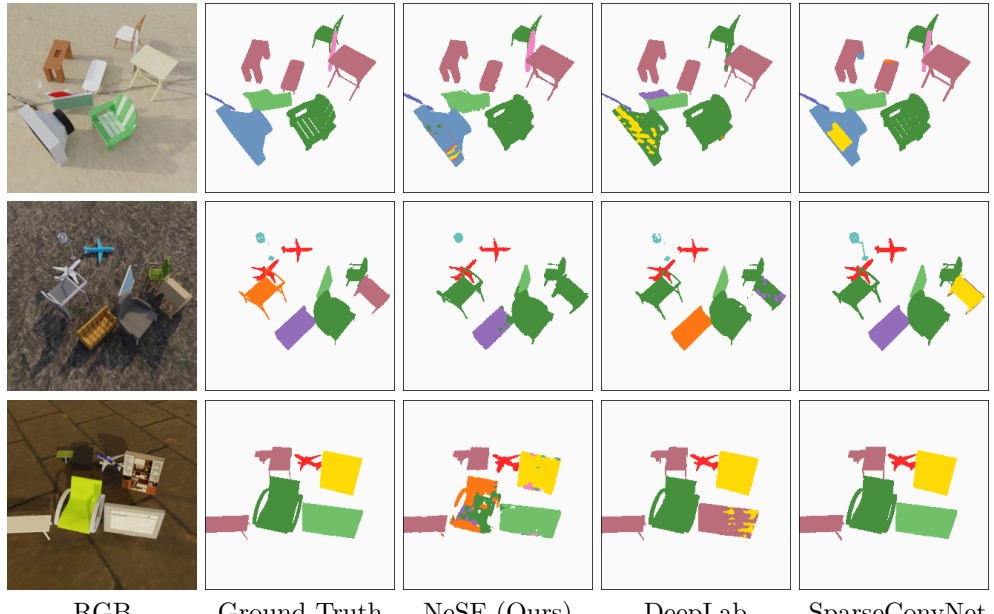

| RGB | Ground Truth | NeSF (Ours) | DeepLab | SparseConvNet |

Figure 5: **Qualitative comparison (`ToyBox13`)** – Unlike DeepLab, NeSF is able to clearly separate objects with similar appearance but different geometry (Top). However, NeSF struggles with thin structures like lamp posts (Middle) and tends to smear labels from nearby objects (Bottom). SparseConvNet suffers from neither limitation but has access to oracle 3D geometry and full 3D supervision.

in the presence of certain dataset shifts. Accordingly, we compare NeSF to Semantic-NeRF on the following perturbations of the `KLEVR` dataset that contain a significant shift in color or texture properties between train and test objects:

- **Mono-Color**: TRAIN SCENES contain monochrome-colored objects, whereas NOVEL SCENES contain randomly sampled colors for each object.
- **Fixed-Color**: Each object with the same semantic category has the same RGB color assigned to it within TRAIN SCENES. Within NOVEL SCENES, each object is assigned a random color.
- **Color-Color**: Each object within the same semantic category is randomly assigned a color from a fixed set of colors at TRAIN SCENES. At NOVEL SCENES, a different set of colors is used for each semantic category.

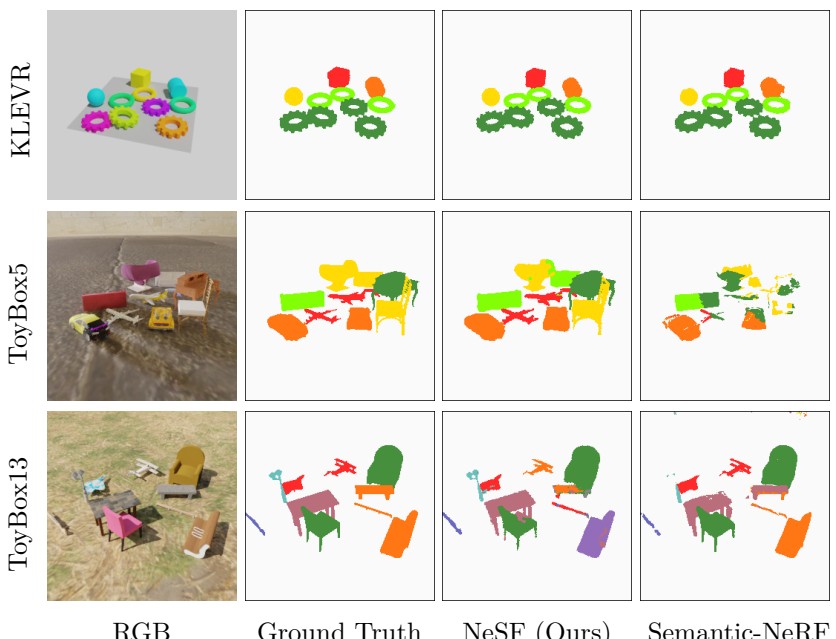

Figure 6: **Qualitative comparison on KLEVR, ToyBox5 and ToyBox13** – On the KLEVR dataset, NeSF performs similar to Semantic-NeRF. On both versions of the ToyBox dataset however, we can see that NeSF is better able to capture fine structures compared to Semantic-NeRF.

- **Texture-Texture**: Each object within the same category is randomly assigned a texture from a fixed set of textures at TRAIN SCENES. Within NOVEL SCENES, a new texture is used for each semantic category.

Figure 7 shows example images from each of the above perturbation sets.

We believe that NeSF's generalization capabilities are a direct consequence of utilizing an architecture that relies on 3D geometry for learning semantics. Meanwhile, Semantic-NeRF generalizes primarily across 2D image space. Thus, we expect the latter to under-perform given changes in the dataset domain caused by varying color and texture.

We follow the same procedure for training our NeSF model as mentioned in 5.1. Similar to previous experiments, we train two variants of the Semantic-NeRF model; using ground-truth labels and pseudo labels from a pre-trained DeepLab model that is fine-tuned on the TRAIN SCENES.

On each of the four datasets, we observe NeSF outperforms Semantic-NeRF as seen in Table 4. On the Mono-Color dataset shift, our method outperforms by 71% and 64% in 2D and 3D mIoU respectively. On the Fixed-Color dataset shift, we observe a difference of 54.5% and 50.4%. Semantic-NeRF's best performance is achieved on the Color-Color dataset shift, where compared to NeSF, we observe a small difference of 3.8% and 2.7% in 2D and 3D mIoU respectively. In the Texture-Texture setting, NeSF outperforms Semantic-NeRF by 52% and 48.8% in 2D and 3D mIoU respectively.

These experiments demonstrate the potential drawbacks of relying primarily on fusing features from 2D for achieving generalization. On the other hand, leveraging the scene's 3D geometry to learn semantics as proposed in the current model, renders NeSF robust to several types of visual dataset shift.

As the proposed dataset perturbations are challenging for 2D semantic methods, the errors made by DeepLab propagate to Semantic-NeRF's predictions. This can be seen in the Fixed-Color and Mono-Color results in Figure 8, where Semantic-NeRF is unable to identify simple objects effectively. This failure mode tracks on even more subtle dataset shifts such as the Color-Color dataset, where we observe an entire object incorrectly segmented by Semantic-NeRF. Conversely, in all cases, NeSF is able to handle the dataset shift seamlessly as demonstrated by both quantitative and qualitative results. In this work we have explored dataset shift in the

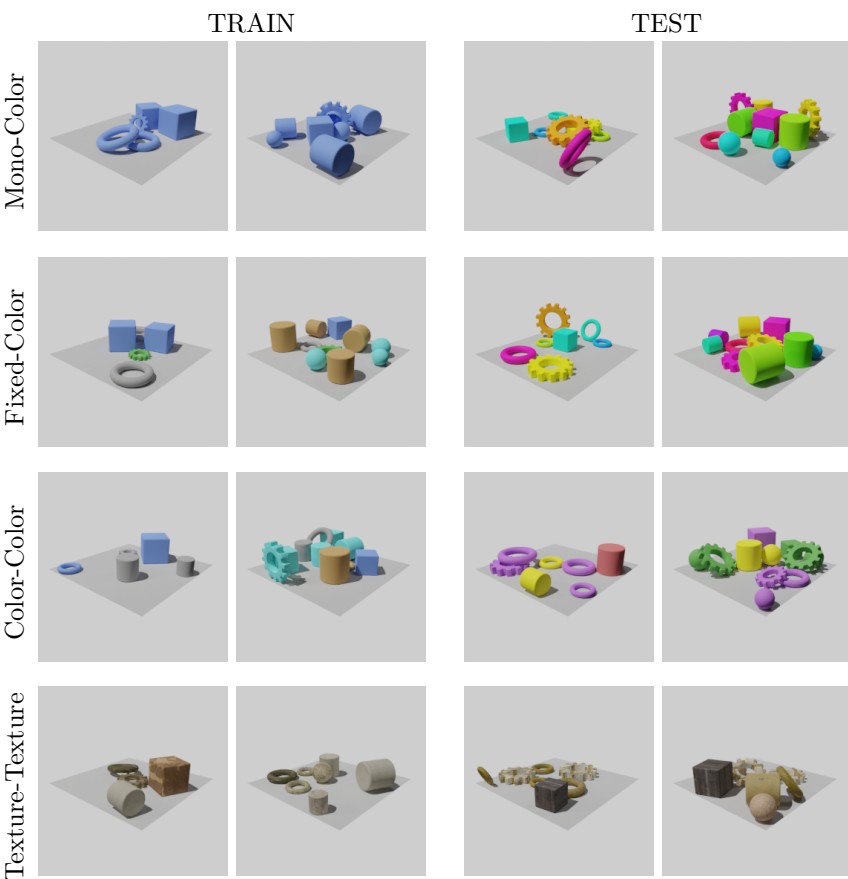

Figure 7: **KLEVR Color Perturbations** – Various perturbations applied to generate color based shifts between train and test scenes.

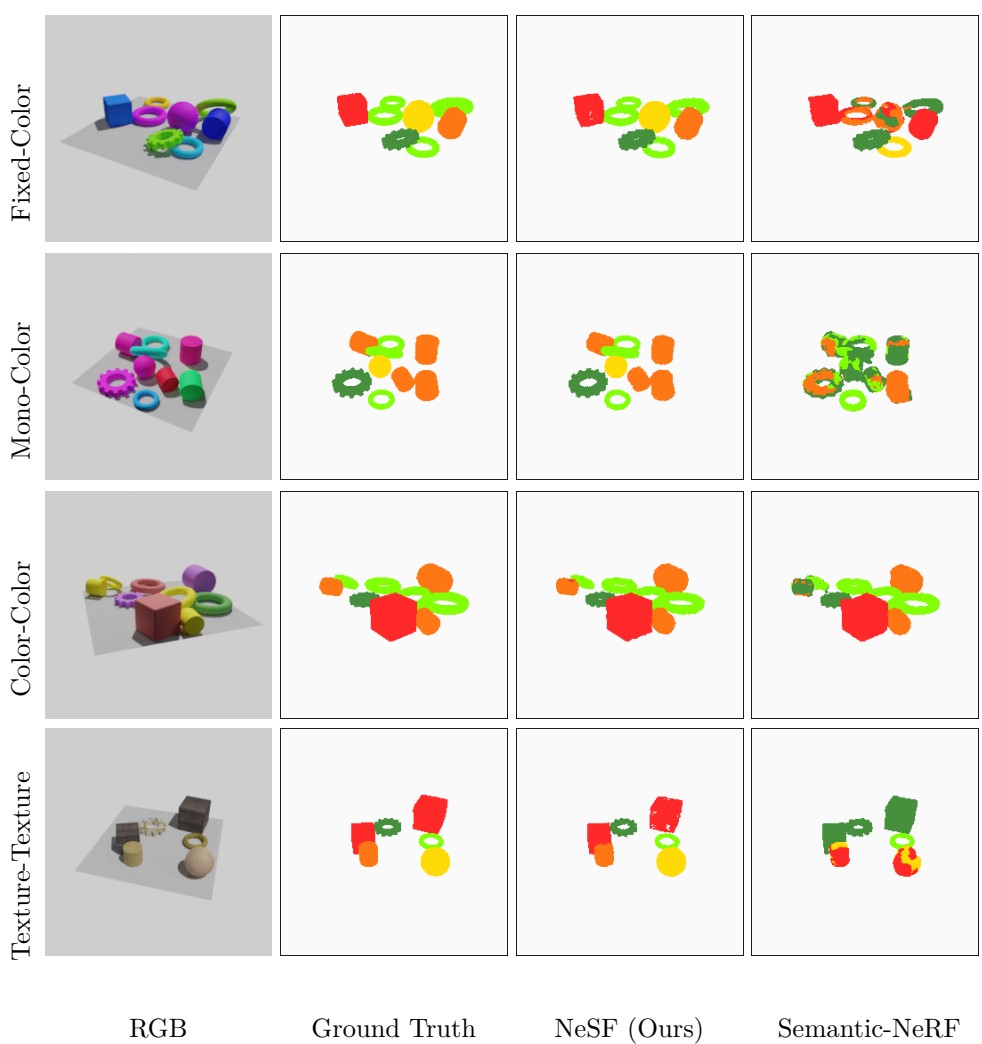

Figure 8: **Qualitative comparison on different dataset pertubations** – By utilizing the shape information to predict the semantic category, NeSF is more robust to shifts in the data domain.

context of visual texture. However, another common form of dataset shift includes inference on data with object categories unseen during training; namely, zero-shot segmentation. Though not the focus of our work, this task is a broad and important area of study. We anticipate the same previous challenges observed and solutions developed for 2D segmentation tasks (Bucher et al., 2019) would extend to Semantic-NeRF and NeSF as well.

## 6 Considerations for Societal Impact

**Compute**. A crucial component of the proposed method is training of a NeRF per scene in order to produce a scene representation. As our training scheme utilizes 32 cores per NeRF training run, the parallelization of training for hundreds of scenes can require thousands of cores to achieve the efficiency we have reported. Though the total number of train cycles is relatively short for a single NeRF (25,000 steps in 23 minutes), there is still the negative impact on the environment to consider in the face of iterative experiments. Additionally, the amount of compute required to efficiently train the representations needed, may limit the scope of researchers that can readily build on our method. In order to tackle the latter aspect, we will release pre-trained NeRF models for all of the scenes we provide within this work in an attempt to make the work more accessible to the broader research community.

**Privacy**. We enable 3D models to be trained without the need for 3D supervision; opening up the possibility of running 3D inference on an entirely new regime of datasets. 3D models of an environment can be obtained without the need of LIDAR sensors or 3D labelling of the data. While ground-breaking, it opens up the possibility for unforeseen use cases for example tied to surveillance data which can be easily collected in 2D, resulting in unintended negative consequences to the privacy of the general public.

## 7 Conclusions and Limitations

In this work, we present NeSF, a novel method for simultaneous 3D scene reconstruction and semantic segmentation from posed 2D images. Building on NeRF, our method is trained solely on posed 2D RGB images and semantic maps. At inference time, our method constructs a dense semantic segmentation field that can be queried directly in 3D or used to render 2D semantic maps from novel camera poses. We compare NeSF to competitive baselines in 2D and 3D semantic segmentation on three novel datasets.

In more challenging settings, we find that NeSF underperforms competitive 2D and 3D methods but outperforms the comparable Semantic-NeRF baseline. Relative to all alternative methods, NeSF offers novel capabilities. Unlike traditional 2D segmentation methods, NeSF fuses information across multiple independent views and renders semantic maps from novel poses. Unlike 3D point cloud methods, NeSF operates on posed 2D information alone at both train and test time. Unlike Semantic-NeRF, NeSF leverages the 3D geometry of neural fields to learn semantics, enabling robustness to visual dataset shifts in color or texture. One limitation of the current architecture is the discritization of the density field into a grid. On the current set of datasets, NeSF struggles to capture small-scaled objects. We anticipate that applying NeSF in the current format to large scale scenes would be computationally inefficient due to the large empty spaces commonly present in such areas. We chose to limit NeSF to a core set of features to better explore the fundamental trade-offs and capabilities of such an approach. We further explore trade offs with respect to model choice in the supplementary material.

In addition to NeSF, we propose three new datasets for multiview 3D reconstruction and semantic segmentation totalling over 3,000,000 frames and 1,000 scenes. Each dataset contains hundreds of scenes, each consisting of a set of randomly placed objects. The more challenging of these datasets are rendered with realistic illumination and a large catalogue of objects and backgrounds. These datasets along with accompanying code and pretrained NeRF models, are publicly available on our project website (to be linked upon publication).

In future work, inspired by competitive 2D and 3D baselines, we anticipate extending NeSF to incorporate 2D semantics; alike Semantic-NeRF and 3D sparsity; alike SparseConvNet will further improve accuracy. Moreover, we intend to investigate the effects of utilizing a different backbone to NeSF, with the goal of building off the improvements in the literature to tackle some of the limitations encountered by our model such

as lack of ability to handle motion blur (Ma et al., 2021), reducing floater artifacts and modeling unbounded scenes (Barron et al., 2022), and the high computational cost of training NeRF models (Müller et al., 2022). While we evaluate NeSF on synthetic data due to the lack of well-suited large multi-scene real-world NeRF datasets, recent research has started tackling these problems (Zhang et al., 2022; Tancik et al., 2022), further motivating the development of techniques for semantic segmentation of neural fields alike NeSF.

## Acknowledgements

We would like to express our gratitude to Konstantinos Rematas, D. Sculley, and especially Thomas Funkhouser for their ideas and suggestions. We would like to note our deep appreciation for the project support and leadership given by Jakob Uszkoreit, without which the work would not have been possible. The authors would also like to thank Rocky Cai in his assistance in assembling the DeepLab baseline experiments.

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

# NeSF: Neural Semantic Fields
# for Generalizable Semantic Segmentation of 3D Scenes

## Appendix

## A  Training Details

We describe the model architecture and training procedure used by NeSF and its baselines below. Unless otherwise stated, we train each method using all TRAIN SCENES with 9 randomly-selected images per scene. For KLEVR, this results in 100 TRAIN SCENES; for ToyBox5 and ToyBox13 500 TRAIN SCENES each. All methods are evaluated on 4 randomly-selected images from each dataset's NOVEL SCENES. For KLEVR, this results in 20 NOVEL SCENES; for ToyBox5 and ToyBox13, 25 NOVEL SCENES each. We ensure that each method observes the same randomly-selected set of images per scene by specifying the seed of the random number generator.

**NeRF**. The first stage of NeSF is the training of per-scene NeRF models. We employ the model architecture and training regime of Mildenhall et al. (2020). Each scene's density field is described by an MLP with 8 hidden layers of 256 units, and its appearance by an additional MLP of 1 hidden layer and 128 units. We employ 7 octaves for positional encoding. Each NeRF model is trained on pixels selected at random from 9 views with the Adam optimizer (Kingma & Ba, 2014). The learning rate is exponentially decayed from $1e-3$ to $5.4e-4$ over 25,000 steps. We train each NeRF model for approximately 20 minutes on 8 TPUv2 cores.

**NeSF**. NeSF has two major model components: a 3D UNet and an MLP Decoder. For the 3D UNet, we employ the UNet architecture of Çiçek et al. (2016) with the BatchNorm layers removed and only 2 max-pooling operations. We use 32, 64, and 128 output channels prior to each max-pooling operation. For the MLP Decoder, we employ 2 hidden layers of 128 hidden units each with a ReLU non-linearity.

We train NeSF with Adam optimizer (Kingma & Ba, 2014). We use an exponentially decaying learning rate initialized to $1e-3$ and decaying to $1e-5$ over 25,000 steps. At each step, we employ a stratified sampling approach: we randomly select 32 scenes, then randomly select a set of 128 pixels from each scene's TRAIN CAMERAS. For volumetric rendering, we sample 192 points along each ray according to the stratified approach employed in NeRF (Mildenhall et al., 2020). For each scene in the batch, we discretize NeRF's density grid by probing at $64^3$ evenly-spaced points. Before discretizing, we apply a random rotation about the z-axis (upwards) to each scene. For smoothness regularization, we uniformly sample 8,192 additional 3D coordinates from each scene and add random noise with standard deviation 0.05. When computing the loss, we assign a weight of 0.1 to the smoothness regularization term.

We find that we are able to train NeSF to convergence in approximately 45 minutes on 32 TPUv3 cores.

**Semantic-NeRF**. We implement Semantic-NeRF following the architecture design of Zhi et al. (2021). The architecture follows the same design as the NeRF architecture previously described with the addition of an MLP with 2 hidden layers of 128 units for modeling the semantics. We use 7 octaves for positional encoding and follow the stratified sampling approach employed in NeRF (Mildenhall et al., 2020) with a total of 192 samples (64 coarse and 128 fine). We train with Adam optimizer (Kingma & Ba, 2014), using an initial learning rate of 1e-3 that is decayed exponentially to 5.4e-4 over the course of 25,000 steps. We train a single Semantic-NeRF model per-scene for the NOVEL SCENES. Each Semantic-NeRF model trains for approximately 20 minutes on 8 TPUv2 cores.

**DeepLab**. We train a DeepLab Wide-ResNet-38 model (Wu et al., 2019), warm starting with a checkpoint pre-trained on COCO. For our optimization scheme, we apply SGD + Momentum with a slow start learning rate of 1e-4 and a linear ramp up to 6e-3 followed by a cosine schedule decay in learning rate to 1.26e-7 at 55,000 training steps. We additionally apply weight decay of $1.0e-4$. For each train step, we use a batch size of 32. Models are trained on 32 TPUv3 chips. To enable re-use of a well-performing hyperparameter configuration, we up-sample our input images from 256x256 to 1024x1024, using bilinear interpolation for the RGB input and nearest neighbor interpolation for the corresponding semantic maps.

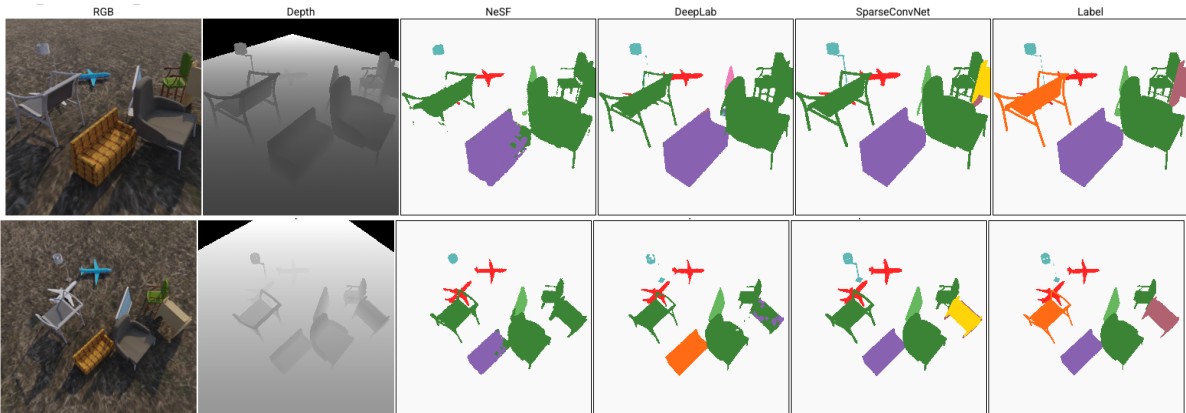

Figure 9: **Multiview Consistency** While NeSF and SparseConvNet classify the orange couch and the display identically from multiple independent views of the same scene, DeepLab's predictions vary.

**SparseConvNet**. Our SparseConvNet (Graham et al., 2018) implementation is based on the TF3D (Huang et al., 2020) and 2D3DNet (Genova et al., 2021) implementations. Each convolutional layer except the last is an occupancy-normalized 3x3x3 sparse spatial convolution followed by batch norm and then ReLU. The final layer omits batch norm and ReLU. Each encoder stage is a pair of convolution layers followed by a 2x2x2 spatial max-pool operation, and each decoder layer is a voxel unpooling operation followed by a pair of convolutional layers. The encoder feature widths are (64, 64), (64, 96), (96, 128), (128, 160), (160, 192), (192, 224), (224, 256). These are the output channel counts of the first and second convolutional layers per block. The bottleneck is a sequence of two convolutional layers of widths 256 each. The decoder feature widths are (256, 256), (224, 224), (192, 192), (160, 160), (128, 128), (96, 96), (64, 64). Finally we apply a sequence of three convolutional layers with sizes (64, 64, `class_count`), which are followed by a softmax layer and a cross-entropy loss function. Our input features are only occupancy (i.e., a 1 on all input points). We use 0.005-width voxels in a [-1, 1] cube scene. We optimize for 450,000 steps with SGD using a momentum of 0.9, a batch size of 5, an initial learning rate of 0.015, and a cosine learning rate decay starting at step 200,000 and ending at step 450,000. We add an $\ell_2$ weight decay loss of $1e-4$ and train asynchronously on 20 NVIDIA V100 GPUs. We apply the following data augmentations: XY rotations of up to $\pm 10$ degrees, z rotations of $\pm 180$ degrees, and a random scale factor between 0.9 and 1.1.

# B   Analysis

## B.1   Qualitative Results

In Figures 14, 15, and 16, we present randomly-selected qualitative results on each dataset studied in this paper. In each row, we depict the ground truth RGB, depth, and semantic map alongside 2D segmentation maps produced by NeSF, DeepLab, and SparseConvNet. We observe that all methods are effective at separating foreground objects from the floor and background. Unlike SparseConvNet, NeSF and DeepLab tend to assign different parts of the same object to different semantic categories when the correct category is ambiguous.

While NeSF and SparseConvNet are multi-view consistent by design, this is not the case for 2D methods such as DeepLab. In Figure 9, we demonstrate one instance of 3D inconsistency. In this example, NeSF and SparseConvNet label the orange couch and white-blue display identically from both views, whereas DeepLab's classification changes.

**NeSF 3D Density Field Quality**. Notably, a key difference between SparseConvNet and NeSF is the provision of a ground truth point cloud as input for SparseConvNet. Several aspects of SparseConvNet may contribute to its overall superior performance relative to NeSF including access to oracle 3D geometry, sparse point cloud input representation, or the SparseConvNet model architecture. To better understand where headroom exists for improvement of NeSF, we begin by visually inspecting the difference in 3D geometry

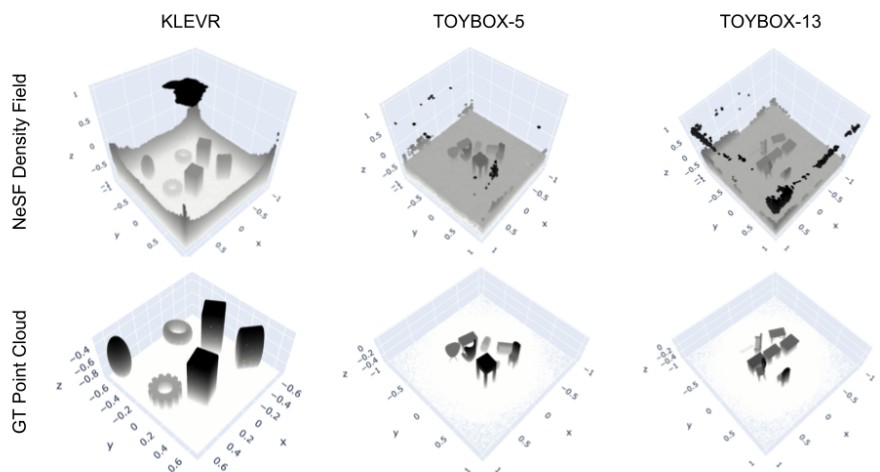

Figure 10: **NeRF 3D Density Field (top) and Ground Truth Point Clouds (bottom) for `KLEVR`, `ToyBox5`, and `ToyBox13`.** We find that NeRF's density field accurately captures the 3D geometry of the scene. The NeSF density fields are sampled at a resolution of 128x128x128, and are filtered for ease of visualization for positive $\sigma$ values with thresholds of 16, 64, and 64 for the 3 datasets respectively.

Table 7: **Ablation: hyper-parameters** – Data augmentation, in the form of random scene rotations, increased spatial resolution of the density grid, and increased UNet model capacity improve 2D and 3D mIoU. Experiments on 500 scenes from the `ToyBox5` dataset.

| Hyperparameter | | 2D | 3D |
|---|---|---|---|
| Random Rotations | No | 69.5 | 83.6 |
| | Yes | 78.8 | 89.7 |
| Density Grid | (32, 32, 32) | 71.1 | 81.5 |
| | (48, 48, 48) | 76.4 | 89.3 |
| | (64, 64, 64) | 78.8 | 89.7 |
| UNet | (16, 32, 64) | 80.6 | 89.1 |
| | (24, 48, 96) | 80.1 | 89.8 |
| | (32, 64, 128) | 79.0 | 89.8 |
| MLP | (0, 32) | 78.6 | 90.7 |
| | (1, 32) | 79.7 | 89.8 |
| | (1, 64) | 80.7 | 89.4 |
| | (2, 128) | 79.2 | 89.5 |

between the 3D density field of NeSF in relation to the ground truth point cloud provided to SparseConvNet for scenes selected from `KLEVR`, `ToyBox5`, and `ToyBox13` in Figure 10. We observe the NeSF density fields often miss thin structures and fine details, and "floaters" are particularly evident in `ToyBox13`. Improvement of density field quality via improvements to NeRF representations may resolve such artifacts. Furthermore, akin to results in Table 8, such improvements would likely improve the performance of NeSF. We leave additional inspection of 3D input representation and replacement of the semantic model architecture of NeSF for future work.

## B.2  Ablations

**Model Ablations – Table 7**. We repeat our our ablation study on the `ToyBox5` model and observe overall consistent results with `KLEVR` model ablations. Table 7 shows results varying each component. Similar to the results on `KLEVR`, we observe that data augmentation in the form of random scene rotations improves quality the most, adding 9.3% and 6.1% to 2D and 3D mIoU respectively. The spatial resolution of the probed NeRF density grids is again confirmed as crucial, and notably to a greater extent than for `KLEVR`. We hypothesize that this occurs as `ToyBox5` contains more fine structured objects than `KLEVR`.

Table 8: **Ablation: sensitivity to reconstruction quality for `ToyBox5`** – The accuracy of our method improves with NeRF's reconstruction quality. PSNR and SSIM are are averaged across all scenes and metrics aggregated. Experiments on all scenes from `ToyBox5`.

| # RGB | NeRF | | NeSF | |
|---|---|---|---|---|
| Images | PSNR | SSIM | 2D | 3D |
| 5 | $17.5 \pm 2.1$ | $0.55 \pm 0.15$ | 15.0 | 17.9 |
| 10 | $19.2 \pm 2.9$ | $0.62 \pm 0.15$ | 29.1 | 35.2 |
| 25 | $23.9 \pm 2.7$ | $0.76 \pm 0.09$ | 61.4 | 74.1 |
| 50 | $26.3 \pm 2.1$ | $0.81 \pm 0.06$ | 72.3 | 88.7 |
| 75 | $27.3 \pm 2.0$ | $0.83 \pm 0.05$ | 72.6 | 89.5 |
| 100 | $27.9 \pm 2.0$ | $0.84 \pm 0.04$ | 73.6 | 90.0 |

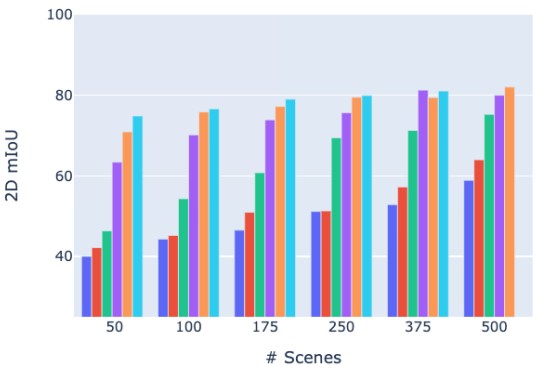 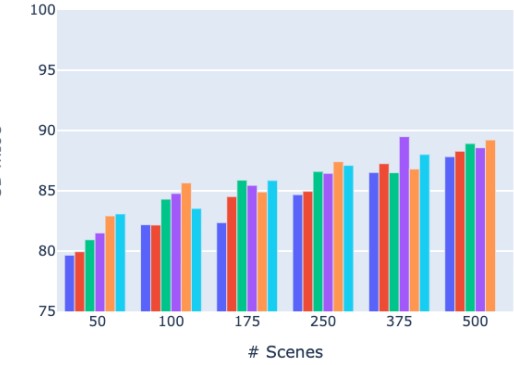

Figure 11: **Ablation: data efficiency** – 2D and 3D mIoU as a function of the number of train scenes for scenes with supervision from **1**, **2**, **5**, **10**, **25**, or **50** semantic maps per scene. Note, **50** semantic maps did not fit in memory for 500 scenes, hence this particular setup is dropped from the results. NeSF generalizes to new scenes with as few as a *one semantic map per scene*. Additional semantic maps per scene marginally improve the accuracy. Experiments on `ToyBox5` dataset.

**Sensitivity to reconstruction quality** – **Table 8**. We re-evaluate the robustness of NeSF to NeRF reconstruction quality in the context of the `ToyBox5` dataset. To modulate reconstruction quality, we vary the number of RGB images used when fitting NeRF models from 5 to 100 and confirm NeRF reconstruction quality improves as more RGB images are provided. As previously observed, the 2D and 3D segmentation quality of NeSF improves monotonically with NeRF's reconstruction quality. Notably, 3D segmentation accuracy begins to level off near 88% when NeRF models are optimized with as few as 50 RGB images, with a large jump in performance between 25 and 50 images.

**Sensitivity to data scarcity** – **Figure 11** We repeat our analysis providing limited numbers of semantically labelled maps for NeSF's training on the `ToyBox5` model. We vary the number of provided label maps from 1 to 50. Similar to the `KLEVR` setup, we observe that providing additional semantic maps per scene improves the performance, with a large jump between 5 and 10 maps. A saturation in the model performance is reached at around 25 maps per scene. Moreover, the model is still able to generalize with as little as 1 semantic map per scene.

**Comparison to NeRF SparseConvNet** – **Table 9** To better understand the performance gap between SparseConvNet and NeRF, we replace SparseConvNet's oracle point cloud with one derived from NeRF's density field. In particular, we derive point clouds from NeRF by sampling the density field on a grid, thresholding to only include points above a heuristically chosen density value ($\sigma = 64$), and removing all

Table 9: **SparseConvNet Ablation** – NeSF is compared to SparseConvNet models trained on NeRF-derived and oracle point clouds on the `ToyBox13` dataset.

| | Point Cloud | 3D mIoU |
|---|---|---|
| NeSF | – | 0.62 |
| SparseConvNet | NeRF | 0.68 |
| SparseConvNet | Oracle | 0.82 |

|  | ToyBox5 | | | | ToyBox13 | | | |
|  | TRAIN CAMERAS | | TEST CAMERAS | | TRAIN CAMERAS | | TEST CAMERAS | |
|  | 2D mIoU | 3D mIoU | 2D mIoU | 3D mIoU | 2D mIoU | 3D mIoU | 2D mIoU | 3D mIoU |
|---|---|---|---|---|---|---|---|---|
| NeSF | $79.5 \pm 1.3$ | $89.1 \pm 0.9$ | $78.8 \pm 1.4$ | $90.2 \pm 0.8$ | $54.3 \pm 2.0$ | $58.7 \pm 1.7$ | $54.0 \pm 2.2$ | $60.0 \pm 2.6$ |
| NeSF(+rgb features) | $80.0 \pm 1.3$ | $88.8 \pm 0.8$ | $79.8 \pm 0.8$ | $89.1 \pm 1.1$ | $54.7 \pm 0.9$ | $59.2 \pm 0.9$ | $54.2 \pm 1.5$ | $60.1 \pm 1.3$ |

Table 10: **NeSF with RGB Features** – NeSF with RGB Features performance relative to NeSF. We extend geometric density features to include RGB features sampled at an arbitrary but consistent viewing direction. The features are attenuated by the respective positional density of the field at their location in the grid, and NeSF is trained with the extended set of features. We do not observe a significant difference in performance given the particular datasets used for evaluation.

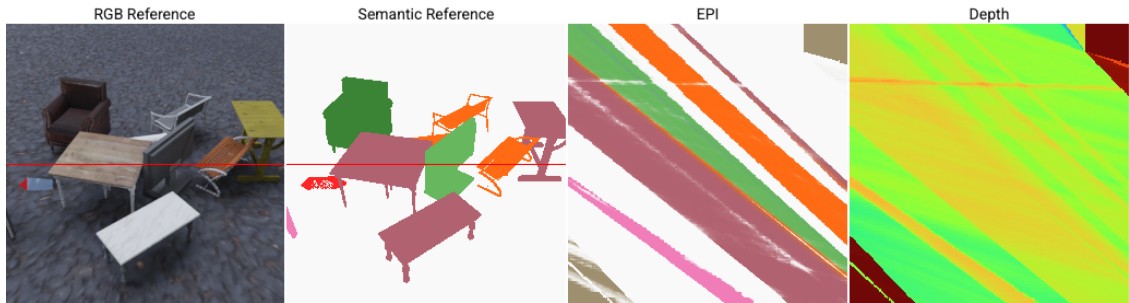

Figure 12: **Epipolar Plane** We demonstrate the 3D consistency of NeSF by rendering the epipolar plan along the red scan line as the camera moves from right to left. The epipolar plane is is smooth and consistent except when a "floater" passes in front of the camera.

points within the dense floor below a cutoff height (z=-0.55). Semantic labels are provided by transferring from the nearest neighbor point in the respective oracle point cloud. The labeled point clouds are used to train a SparseConvNet model, and inference is performed on unlabelled NeRF point clouds for test scenes. For evaluation, mIoU is calculated by comparing oracle point cloud labels against the nearest neighbor label per point from the predicted labels for NeRF point cloud test scenes. Though still a slightly unfair comparison due to the use of ground-truth 3D data for the purpose of labeling the NeRF point clouds, we find the experiment informative; demonstrating the far smaller gap between NeRF-SparseConvNet and NeSF performance. The smaller performance gap indicates the initial significant performance difference may be largely attributed to imperfect geometric reconstructions, ill-defined density within objects, and/or floater artifacts resulting from NeRF based geometry reconstruction. Due to extra time and resource overhead for training SparseConvNet, we limit the experiment to our most challenging dataset `ToyBox13`.

**Addition of RGB features** – **Table 10** We explore the utility of extending the density field input to the UNet of NeSF to additionally contain the RGB features from the radiance field of the NeRF. To sample RGB features along a spatial 3D grid, viewing directions along with positions must be provided to the NeRF MLP. While possible to sample radiance along a calculated surface normal, NeRF geometry is known to be foggy as opposed to being tightly concentrated at surfaces, making normal vectors quite noisy when calculated (Verbin et al., 2021); moreover, such normals are meaningless in empty space as well as within objects. As an alternative for our datasets specifically, we consider that the ShapeNet objects utilized for the `ToyBox5` and `ToyBox13` datasets are composed of diffuse materials, and radiance can be considered independent of viewing direction. As such, when sampling the radiance features along the 3D spatial grid, we simply sample at an arbitrary viewing direction (i.e. (1,1,1)). To ensure features are representative of filled space, we attenuate the RGB features by the density field, and concatenate the density as well. In our experiments we observe performance remains essentially on par with the use of density alone, but suspect results may be specifically due to a lack of strong correlation between object color/texture and class in our datasets.

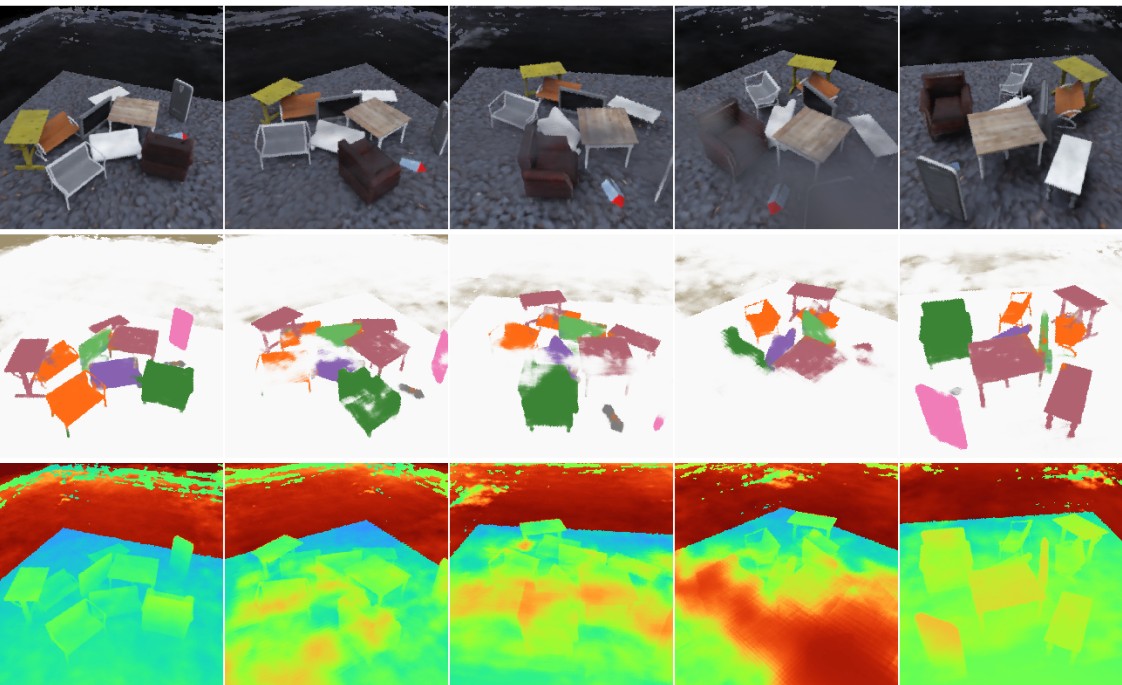

Figure 13: **Floaters** – In the above 3 rows, we illustrate NeRF's RGB reconstruction (top), NeSF's semantic field, and NeRF's density field (bottom) over the course of 5 video frames. When NeRF's density field contains "floaters", NeSF often assigns them to the background semantic category. Note that recent methods such as Mip-NeRF-360 (Barron et al., 2022) have recently introduced additional losses to remove such artifacts.

### B.3 Multiview Consistency

Unlike conventional 2D methods, NeSF is 3D-consistent by design. In Figure 12, we visualize the epipolar plane traced out along the red reference line for NeSF's semantic predictions. We find that the resulting predictions are consistent and smooth except when a floater obstructs the camera's view, as illustrated by the white smudge in the EPI. In Figure 13, we see a further example of this phenomena. As the camera rotates about the scene, a floating mass of density obstructs the camera's view, and the resulting semantic maps contain a large number of mislabeled pixels. **We strongly encourage the reader to view additional results in the accompanying video for more detail.** In particular, results depicting floaters for ToyBox5 can be viewed from timestamp 7:34.

### B.4 Limitations

**Confusion matrix**. In Table 11 and Table 12, we present NeSF's per-class confusion matrix on the ToyBox13 dataset for 2D pixel and 3D point classification. While NeSF is able to easily identify larger, articulated semantic categories such as cabinet, chair, display, or table (78.0-89.4% 2D, 79.4-93.6% 3D), it struggles to identify object categories for smaller objects such as rifle (56.3% 2D, 75.3% 3D) or geometrically unarticulated objects such as loudspeaker (38.5% 2D, 40.4% 3D). When NeSF confuses foreground object categories, the most common errors are between geometrically-similar classes. For example, benches are often mislabeled as chairs (17.0% 2D, 17.9% 3D) and sofas (26.5% 2D, 27.4% 3D), and loudspeakers are often mislabeled as tables (32.0% 2D, 32.5% 3D).

**Accuracy 2D vs. 3D**. Our experiments indicate that NeSF's accuracy is *higher* in 3D than in 2D. We found this surprising, especially considering the 2D nature of NeSF's semantic supervision. We believe the ultimate cause to be "floaters" in the 3D density field recovered by NeRF. In Table 11, we see that approximately 5-10% of 2D pixels from each semantic category are mislabeled as "background". In contrast, Table 12

demonstrates that the same type of error is made approximately 1% of the time in 3D. The most prominent exception to this is the bench category, whose objects often contain thin structures poorly captured by NeSF.

**Impact of floaters**. In Figure 13, we qualitatively show how "floaters" reduce NeSF's accuracy in image-space. In this set of 5 video frames, we demonstrate a camera path passing in front of a floating cloud. This cloud is assigned to the background semantic category and obscures the foreground objects from the scene. In spite of the *objects* being correctly labeled, the generated semantic maps are incorrect. As a result, NeSF achieves lower 2D mIoU than 3D mIoU as the latter is not hindered by floaters and is corroborated by Tables 11 and 12. We believe that eliminating this failure in geometric construction will significantly improve NeSF's accuracy. Solutions are readily provided by methods building on NeRF (Yu et al., 2021b). **We strongly encourage the reader to view additional results in the accompanying video for more detail.**

| | background | airplane | bench | cabinet | car | chair | display | lamp | loudspeaker | rifle | sofa | table | telephone | vessel |
|---|---|---|---|---|---|---|---|---|---|---|---|---|---|---|
| background | **99.1%** | 0.1% | 0.0% | 0.0% | 0.0% | 0.2% | 0.1% | 0.0% | 0.0% | 0.0% | 0.0% | 0.2% | 0.0% | 0.0% |
| airplane | 12.9% | **69.2%** | 3.2% | 0.1% | 0.0% | 1.9% | 0.0% | 0.3% | 0.0% | 1.4% | 0.0% | 4.6% | 0.0% | 6.3% |
| bench | 9.1% | 0.7% | **42.1%** | 0.1% | 0.0% | 17.0% | 0.0% | 0.2% | 0.0% | 1.2% | 26.5% | 1.7% | 0.0% | 1.4% |
| cabinet | 2.2% | 0.0% | 0.0% | **78.0%** | 0.0% | 0.1% | 2.7% | 0.0% | 12.8% | 0.0% | 0.0% | 2.7% | 1.5% | 0.0% |
| car | 5.0% | 0.1% | 0.1% | 0.0% | **84.6%** | 0.5% | 0.1% | 0.3% | 0.3% | 0.2% | 1.3% | 4.1% | 0.0% | 3.3% |
| chair | 5.8% | 0.0% | 0.9% | 0.0% | 0.0% | **89.4%** | 0.1% | 0.0% | 0.1% | 0.0% | 2.9% | 0.7% | 0.0% | 0.0% |
| display | 9.2% | 0.0% | 0.3% | 0.4% | 0.1% | 1.2% | **83.3%** | 0.0% | 3.7% | 0.0% | 0.2% | 1.3% | 0.2% | 0.0% |
| lamp | 11.1% | 0.0% | 0.0% | 0.0% | 0.0% | 0.5% | 0.0% | **61.4%** | 18.1% | 0.0% | 0.0% | 0.1% | 7.8% | 0.8% |
| loudspeaker | 5.4% | 0.1% | 4.3% | 10.5% | 0.0% | 0.6% | 3.2% | 1.1% | **38.5%** | 0.0% | 2.2% | 32.0% | 1.7% | 0.4% |
| rifle | 30.7% | 1.6% | 2.0% | 0.2% | 0.0% | 0.7% | 3.1% | 0.0% | 0.0% | **56.3%** | 3.0% | 0.7% | 0.0% | 1.7% |
| sofa | 10.0% | 2.3% | 0.6% | 0.0% | 0.0% | 21.6% | 0.0% | 0.1% | 0.1% | 0.0% | **61.6%** | 0.0% | 0.0% | 3.6% |
| table | 5.3% | 0.0% | 1.7% | 3.1% | 0.0% | 4.6% | 0.1% | 3.1% | 1.2% | 0.0% | 0.7% | **79.9%** | 0.0% | 0.3% |
| telephone | 1.7% | 0.0% | 0.0% | 9.3% | 0.0% | 0.2% | 3.7% | 0.0% | 15.7% | 0.0% | 0.0% | 0.3% | **69.0%** | 0.0% |
| vessel | 5.9% | 12.4% | 0.0% | 0.0% | 0.0% | 0.1% | 0.0% | 0.0% | 0.0% | 0.0% | 4.3% | 0.3% | 0.0% | **77.0%** |

Table 11: **Confusion matrix for 2D semantic segmentations by NeSF on ToyBox13.** Each row corresponds to a ground truth label and is normalized to sum to 100%. NeSF's most common errors include confusing similarly-shaped objects and classifying small and thin objects as background. Correct classifications are highlighted in bold.

| | background | airplane | bench | cabinet | car | chair | display | lamp | loudspeaker | rifle | sofa | table | telephone | vessel |
|---|---|---|---|---|---|---|---|---|---|---|---|---|---|---|
| background | **99.4%** | 0.0% | 0.0% | 0.0% | 0.0% | 0.0% | 0.1% | 0.1% | 0.0% | 0.0% | 0.0% | 0.1% | 0.0% | 0.0% |
| airplane | 1.7% | **78.0%** | 3.6% | 0.0% | 0.0% | 1.8% | 0.0% | 0.2% | 0.0% | 2.1% | 0.0% | 4.8% | 0.1% | 7.6% |
| bench | 6.0% | 0.8% | **42.7%** | 0.2% | 0.0% | 17.9% | 0.0% | 0.3% | 0.0% | 1.7% | 27.4% | 1.6% | 0.0% | 1.4% |
| cabinet | 0.4% | 0.0% | 0.0% | **79.4%** | 0.0% | 0.0% | 2.9% | 0.0% | 13.0% | 0.0% | 0.0% | 2.6% | 1.7% | 0.0% |
| car | 0.8% | 0.4% | 0.3% | 0.0% | **86.7%** | 0.6% | 0.0% | 0.2% | 0.4% | 0.3% | 2.1% | 4.0% | 0.0% | 4.3% |
| chair | 1.0% | 0.0% | 1.1% | 0.0% | 0.0% | **93.6%** | 0.1% | 0.1% | 0.1% | 0.1% | 3.1% | 0.8% | 0.0% | 0.0% |
| display | 0.6% | 0.1% | 0.4% | 0.3% | 0.0% | 1.0% | **90.7%** | 0.0% | 4.8% | 0.0% | 0.4% | 1.4% | 0.3% | 0.0% |
| lamp | 2.3% | 0.0% | 0.0% | 0.1% | 0.1% | 0.4% | 0.0% | **66.7%** | 20.7% | 0.0% | 0.0% | 0.1% | 8.8% | 0.9% |
| loudspeaker | 1.1% | 0.1% | 4.4% | 12.5% | 0.1% | 0.4% | 3.0% | 1.2% | **40.4%** | 0.0% | 2.1% | 32.5% | 1.9% | 0.4% |
| rifle | 2.4% | 4.9% | 4.6% | 0.6% | 0.1% | 1.8% | 3.6% | 0.0% | 0.0% | **75.3%** | 0.9% | 0.0% | 0.0% | 2.7% |
| sofa | 0.7% | 2.9% | 0.8% | 0.0% | 0.0% | 22.8% | 0.0% | 0.1% | 0.1% | 0.0% | **68.2%** | 0.0% | 0.0% | 4.3% |
| table | 0.6% | 0.0% | 1.8% | 3.5% | 0.0% | 4.7% | 0.0% | 3.2% | 1.5% | 0.1% | 0.8% | **82.8%** | 0.0% | 0.8% |
| telephone | 0.1% | 0.0% | 0.0% | 10.5% | 0.0% | 0.0% | 4.5% | 0.0% | 15.8% | 0.0% | 0.0% | 0.0% | **69.1%** | 0.0% |
| vessel | 1.1% | 13.2% | 0.0% | 0.0% | 0.0% | 0.0% | 0.0% | 0.0% | 0.0% | 4.4% | 0.6% | 0.0% | 0.0% | **80.7%** |

Table 12: **Confusion matrix for 3D semantic segmentations by NeSF on ToyBox13.** Each row corresponds to a ground truth label and is normalized to sum to 100%. NeSF's most common errors include confusing similarly-shaped objects and classifying small and thin objects as background. Correct classifications are highlighted in bold.

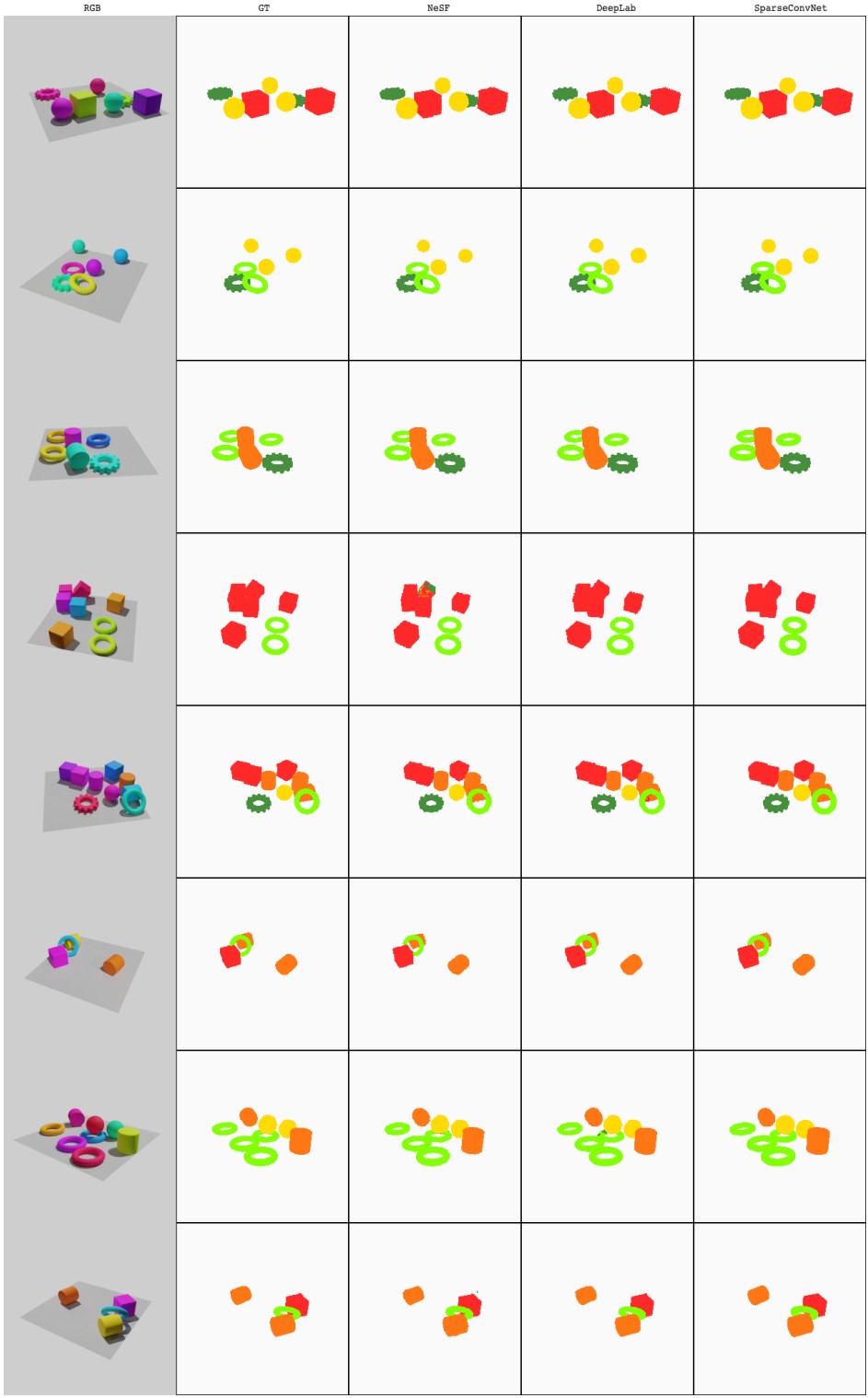

Figure 14: **Additional Qualitative Results on** KLEVR

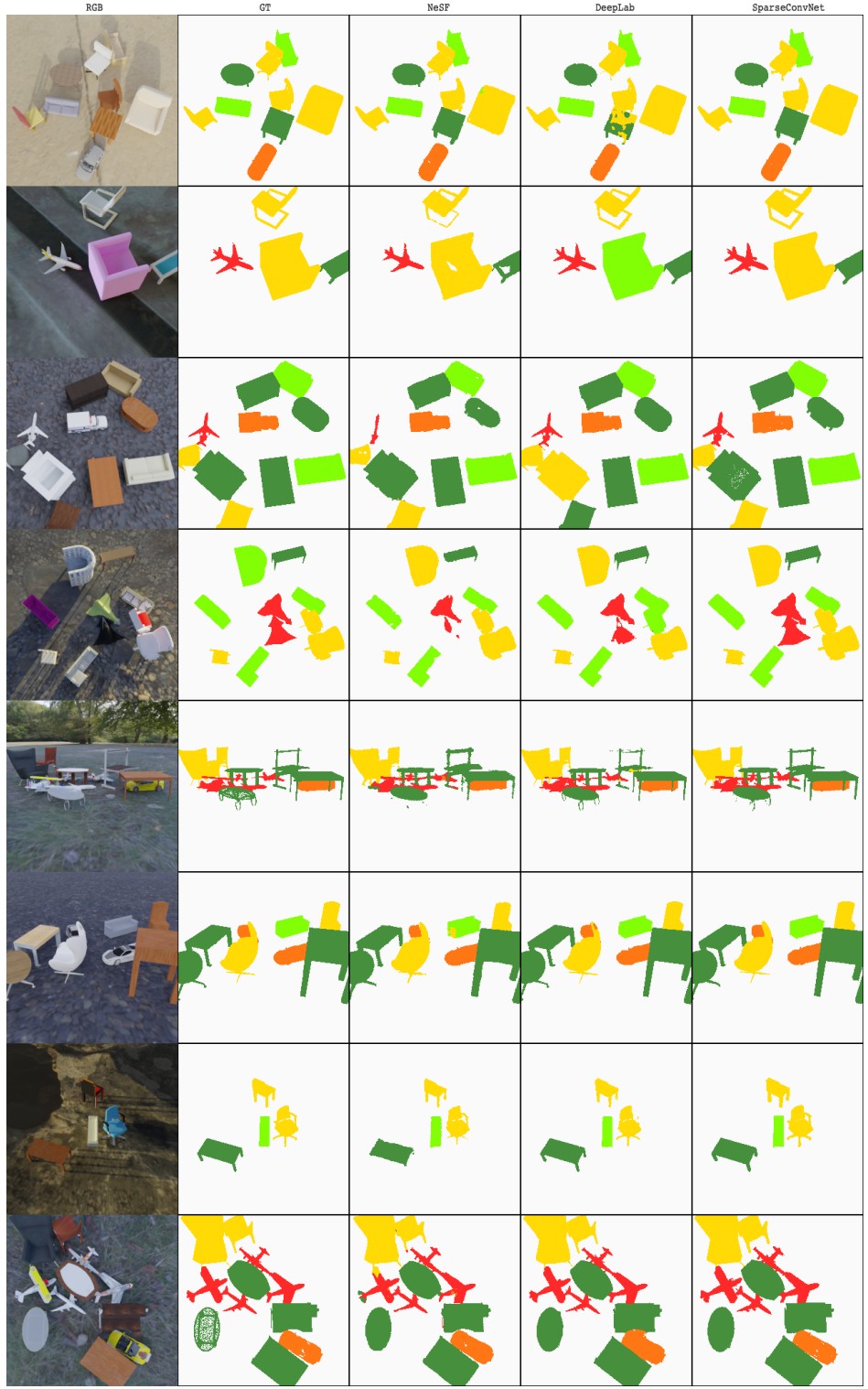

Figure 15: **Additional Qualitative Results on `ToyBox5`**

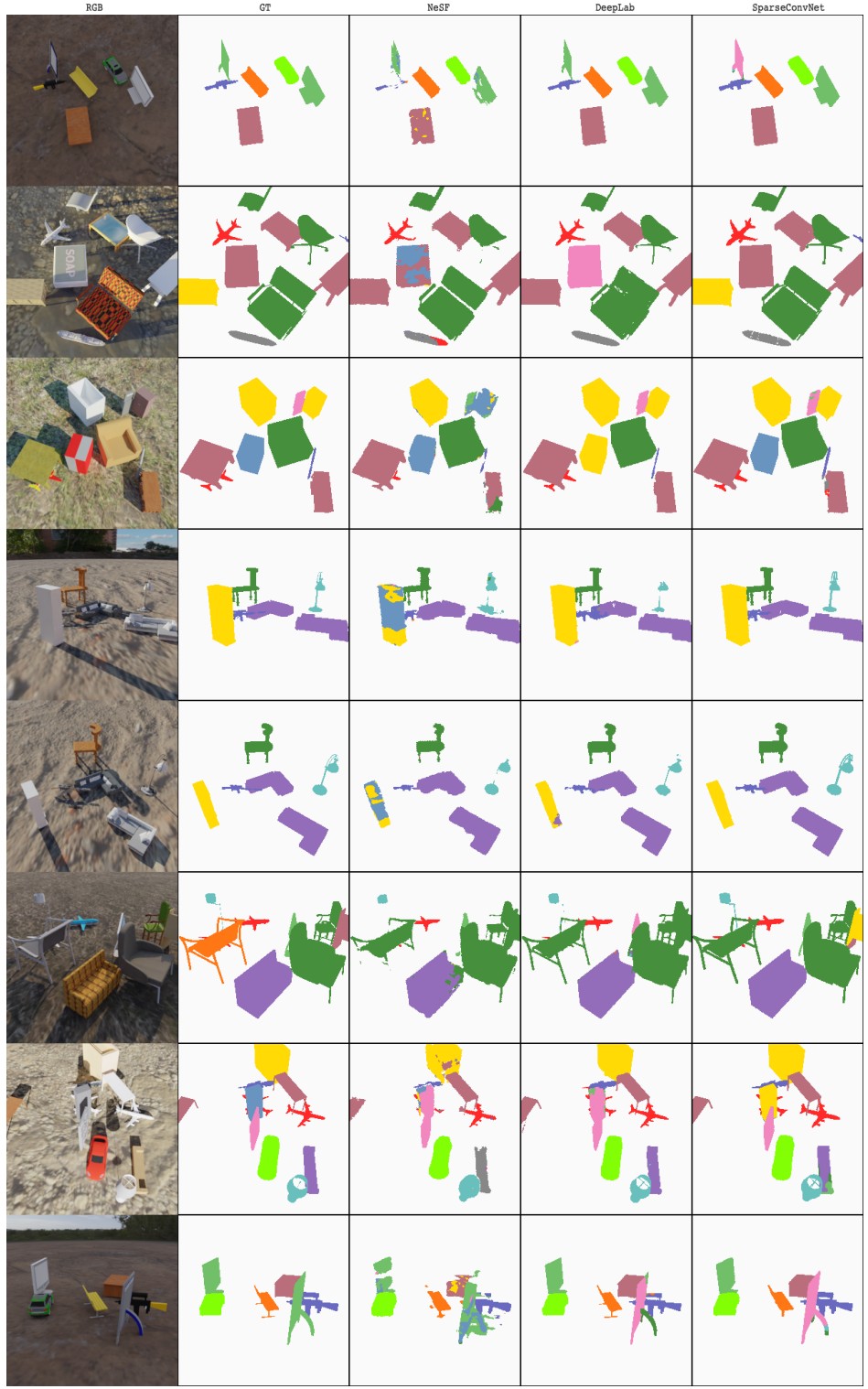

Figure 16: **Additional Qualitative Results on ToyBox13**

