# OpenReview forum: "NeSF: Neural Semantic Fields for Generalizable Semantic Segmentation of 3D Scenes"
_TMLR — Accepted by TMLR_

### Review · Reviewer_u1Zd · 2022-04-27

**Summary Of Contributions:**

This paper presents neural semantic fields -- a class of neural fields that encode scene semantics in addition to the radiance/shape. The method is trained on a collection of NeRF models (pre-)learned from sets of RGB images with corresponding semantic segmentation maps. At inference time, given a NeRF model of a new scene not seen before, the method is able to generalize to produce good semantic fields for new viewpoints. The central technical idea is to operate on the density field of NeRF using a 3D CNN to extract features that can estimate a semantic label. Detailed experiments show that the method outperforms other work on a large corpus of synthetic data.

**Broader Impact Concerns:**

The proposed method is computationally very heavy. Moreover, there are a lot of hyperparameters in the proposed work which could make replication hard. The paper addresses the compute question and proposes to distribute pre-trained NeRF models. I am satisfied that this addresses my replicability concerns.

**Requested Changes:**


- Due to my concerns about computational expense (see broader impact concerns), I would like to see a confirmation from the authors to release pre-trained NeRF models, code, and architecture.

With the above-requested changes, I would be in full support of acceptance.

**Strengths And Weaknesses:**

Overall, I am positive about this paper. The paper has many strengths and some weaknesses that I point out below together with questions, comments, and suggestions for improvement.

### **Strengths**

- This paper takes a step forward in the direction of learning high-level semantics within neural fields -- this is an important and worthy problem to address.
- The insight to operate on the density field is interesting and leads to modularity (NeRF can be pre-trained and distributed).
- It uses only 2D semantic map supervision while lifting it to 3D. It could have broad applications since 2D labels are significantly easier to obtain than 3D labels.
- The paper does a good job, especially in the introduction clearly stating the differences to previous work including NeRF and Semantic NeRF.
- The paper presents a large synthetic dataset of over 1000 scenes.

### **Weaknesses**

- One of the main weaknesses of the paper is the use of synthetic data for experimentation. NeRF took us away from the clutches of synthetic data by showing results on real data. This paper takes us back to synthetic data. While I understand gathering real data at this scale is challenging, it would have also been more compelling -- especially because we have very good tools to extract semantic/instance segmentation masks on real data. Perhaps the authors could provide some more justification for the use of (only) synthetic data.

- In Section 3, the paper mentions that it can be trained with unpaired segmentation maps. But this feature is never demonstrated. The ablations only evaluate scarcity of segmentation maps. It would make the paper stronger if this feature were to be demonstrated.

- As pointed out in the limitations, NeSF relies solely on the density field which is only an indirect function of geometry and contains no information about color. On the other hand, 2D segmentation almost entirely depends only on the appearance. I hypothesize that the approach would work significantly better if appearance information is used.

- The Data Augmentation section is lacking detail. It is unclear how much augmentation is performed and how this affects training times. Table 5 briefly discusses performance but more description would be nice to have.

- In the synthetic data, are views sampled on a fixed radius hemisphere? If so, how does the method generalize to views of different radius?

- It is unclear how the method performs when a new semantic class unseen at training is shown at inference time. What would happen when a new object never seen before appears at inference time?

### **Other Questions/Comments/Suggestions**

- The abstract mentions one-shot learning of semantics -- but it's unclear whether this is at inference time or training time (becomes clear later). Perhaps this could be made more explicit in the introduction?
- How does the choice of epsilon in equation (4) affect the quality of results?
- Page 12, line 4. Fix "??"

---

> ### Author Response · Authors · 2022-05-06
> **Responses**
>
> ### Q1 – “NeRF took us away from the clutches of synthetic data by showing results on real data”
> Please note that the vast majority of NeRF workloads operate in the **overfitting scenarios**, so this statement is true in a very specific setting (single-scene 3D, not multi-scene). Unfortunately, prior real-world datasets have not been compelling to train NeSF (e.g. inadequate views for training NeRFs, nonsubstantive number of scenes, or no labeled semantics). The datasets we considered were ScanNet (issues with motion blur, exposure variation), MatterPort3D (panoramas too sparsely captured), Replica (very few scenes), HyperSim (also synthetic).
>
> We implemented our method with anticipation of additional improvements to NeRF becoming available to mitigate such issues and more appropriate datasets becoming available. As we anticipated, techniques to reliably train on real-world data have now begun appearing (e.g. [mipNeRF360](https://arxiv.org/abs/2111.12077) and [Block-NeRF](https://arxiv.org/abs/2202.05263) which specifically collects data for this purpose. We eagerly await these more suitable datasets being made available to the broader community; once provided with corresponding semantic annotations, we aim to demonstrate NeSF capabilities on real-world datasets.
>
> ### Q2 – “trained with unpaired segmentation maps. But this feature is never demonstrated”
> We apologize for the lack of clarity. We meant unpaired as in many RGB images do not have segmentation associated. Not the other way around, which would clearly not make any sense (how do you ask a user to segment an image w/o RGB available?).
>
> ### Q3 – “I hypothesize that the approach would work significantly better if appearance information is used.”
> A primary goal of our work is to propose a method entirely orthogonal to previous NeRF based semantic segmentation proposals (i.e. SemanticNeRF); namely we focused on producing a method meant to learn from the 3D geometry of the scene in a generalizable manner – a key aspect missing from SemanticNeRF. As a result, the two methods are now neatly complementary and indeed, one could combine SemanticNerf w/ NeSF to further improve performance. We decided to **focus on 3D generalization and its analysis**, and leave the combination of the two approaches to future work. Note however there are very significant advantages to geometry-only approaches; see Figure 8 and Table 9.
>
> ### Q4 – “The Data Augmentation section is lacking detail”
> Briefly, at each step a random rotation of angle [0, 2*pi) is applied, to rotate the density field around the z-axis (up vector). Then, the density grid is constructed and the UNet applied. To query the feature grid, the ray of interest is similarly rotated, in order to sample semantic features appropriately. The remainder of the algorithm remains the same (section 3.2, Data augmentation.) We will add additional details to the camera ready if necessary, but please note that the source code will be publicly released, so the exact design of these augmentations will be readily available. Also note that augmentation does not significantly increase the training time.
>
> ### Q5 – “It is unclear how much augmentation is performed and how this affects training times”
> Both models were trained for up to 5,000 iterations, but appear to converge in <2,000. This corresponds to roughly 10 minutes on a 4x4 TPUv3 configuration
>
> ### Q6 – “How does the method generalize to views of different radius?”
> The field is zero outside the support of the scene, so the behavior would be unchanged. For cameras within the scene, the 3D inductive bias of a NeRF model would ensure that the semantic maps are rendered correctly, as far as the 3D reconstruction is of good quality (hence **your question is more about NeRF than about NeSF?**). Additionally, note our method could be updated to use the recently proposed [mipNeRF360](https://arxiv.org/abs/2111.12077) to deal with unbounded scenes.
>
> ### Q6 – “When a new semantic class unseen at training is shown at inference time.”
> Please note this question could also be asked about the performance of 2D image-to-image networks. Our guess is that the network would get confused. Likely, what would happen is choosing a class for an object with a similar geometric appearance.
> We could run such an experiment if deemed necessary.
>
> ### Q7 – “How does the choice of epsilon in equation (4) affect the quality of results?”
> Please refer to Table 5 (density grid size sweep) to see the effect. Briefly stated, there seems to be moderate but diminishing returns on higher grid resolutions for these datasets, however we anticipate others may benefit further from an increased sampling resolution.
>
> ### Q8 – ”Confirmation from the authors to release pre-trained NeRF models, code, and architecture.”
> Confirmed. Further, note that recent research made efficient training of NeRF model viable to the broader public ([InstantNGP](https://github.com/NVlabs/instant-ngp)).

---

> > ### Comment · Action_Editors · 2022-05-13
> > **Thoughts on authors response?**
> >
> > Hi u1Zd ,
> >
> > Thank you for your review. I wanted to ask whether you have any updated thoughts on the paper given the authors' response? I see that the set of requested changes is quite small (namely confirmation of code/model release) and I think they have been addressed by the authors. But I wanted to nudge discussion along in case there are other things you'd like to discuss with the authors.
> >
> > Thanks,
> >
> > David

---

> > ### Comment · Reviewer_u1Zd · 2022-05-14
> > **Thanks for the response**
> >
> > Thanks to the authors for their response. As I mentioned earlier, I am in support of this work. So my questions and comments were aimed at understanding the work's benefits and limitations better.
> >
> > I am happy with the authors' responses for all except Q1, Q4, and Q6. Details below:
> >
> > ## Q1 – “NeRF took us away from the clutches of synthetic data by showing results on real data”
> >
> > While it is true that most neural radiance field works overfit, *that was not what my comment was about*. My comment was that the results would be more compelling if *some* real data were shown. And as I mentioned, we have very good tools for enabling this: segmentation methods like Mask R-CNN, annotation tools, domain adaptation techniques, etc. The good thing about NeRF is that (unlabeled) data is easy to capture. Even overfitting on 1 scene but showing segmentation from novel views would make the paper more compelling.
> >
> > I would still happily recommend the paper without this experiment, but it would be useful to have a short paragraph explaining the rationale for not doing this. Again, I am not trying to find ways to be negative about the paper (I like it a lot), I am trying to increase its impact. But I get the sense that the paper is committed to the simulator/synthetic data setting... This is fine, but a clarifying para would be nice.
> >
> > ## Q4 – “The Data Augmentation section is lacking detail”
> >
> > Thanks for the data augmentation details. I think it is necessary for these details to be in the paper/appendix. IMO, the paper must be re-implementable without having to dig through dense code to find basic details.
> >
> > ## Q6 – “When a new semantic class unseen at training is shown at inference time.”
> >
> > It is true this question could be asked of any category-level segmentation work -- and I am sure they were asked of 2D segmentation works. It's now time for the NeRF community to ask those questions too. A sentence or two would make it easier for folks to understand the behavior of these models without having to reimplement everything.

---

> > > ### Author Response · Authors · 2022-05-18
> > > **Thanks for the questions**
> > >
> > > ### Q1 – "Even overfitting on 1 scene but showing segmentation from novel views would make the paper more compelling."
> > > You are right, having one/two real scenes would have been an "eye-candy" result in-the-wild, and we simply had not considered it.
> > >
> > > Perhaps the reason for which we did not consider implementing this is that we employed a photorealistic renderer, with realistic illumination maps, and physically based materials. Theoretically, you could replicate one of our 3D scenes in the real-world, hence fully expect **identical** segmentation outcomes to the ones obtained in simulation (clearly, if very carefully executed, especially to ensure the real object has the same scale as the one in the training data, and with the proper $+z$ orientation).
> > >
> > > ### Q4 – "implementable without having to dig through dense code"
> > > Ack, we will add these details to our revisions.
> > >
> > > ### Q6 – "time for the NeRF community to ask those questions too"
> > > Sounds good, we will include this observation in our revisions.

---

### Review · Reviewer_kHpB · 2022-04-28

**Summary Of Contributions:**

This paper presents a method for 3D semantic segmentation without 3D supervision, specifically on the volume density of neural radiance fields (NeRF). Given a dataset of multi-view posed images that allows one to pre-convert to a dataset of NeRF representations, along with the corresponding labeled semantic maps for all or partial viewpoints, the authors advocate to train a 3D U-Net that takes a density grid sampled from the input density field to predict a 3D semantic grid. By volume rendering, the predicted 3D semantic grid can be converted to 2D semantic segmentation predictions. The authors also created 3 datasets with a new rendering engine with different scene complexities to evaluate the proposed method.

**Broader Impact Concerns:**

The potential societal impact was sufficiently addressed in the broader impact statement.

**Requested Changes:**

I would like to see all weaknesses fixed. Aside from the last weakness of referencing, I think the rest are all important to justify for acceptance.

**Strengths And Weaknesses:**

Strengths:
- While the framework is simple, enabling training with one labeled semantic map per scene shows good potential towards 3D understanding with more practical 2D supervisory signals.
- I believe the main contribution of this paper is the use of neural fields for representing 3D data that can be derived from posed 2D images. This has the advantage of applying random geometric data augmentation that samples the density grid in the continuous space while training the 3D ConvNet. The 3D ConvNet and volume rendering (of semantic features) components are rather trivial.
- I think the paper is well-written and the presentation is clear in general. The overall framework is easy to understand, and the experimental settings are described in a detailed manner with sufficient discussions.

Weaknesses:
- The new synthetic datasets are nice for benchmarking the performance of NeSF, but they are inherently a product from the Kubric paper, and I do not think they are considered as a contribution from this paper.
- The comparison against SparseConvNet does not seem to be apple to apple, as the data modality is inherently different. The below baselines/ablations would make the comparison fairer:
  - Use the ground-truth 3D shapes (which effectively has continuous occupancy) to train NeSF, or use the same pretrained NeRF representations, threshold the density to get a surface representation, then sample point clouds for SparseConvNet.
  - Sample the point clouds dense enough to allow differentiable rendering, so that SparseConvNet can be learned from 2D supervision.
  Otherwise, comparing against SparseConvNet under the current setting is pretty meaningless.
- I think some discussions should be made on Fig 4, as it would be expected that mIoU should be strictly improving with the number of supervising semantic maps per scene.
- I think there should at least be some discussions on the failure cases. Since NeSF depends heavily on the performance of NeRF, there should be some discussions on the robustness of NeSF to the quality of NeRF (e.g. when NeRF fails to optimize reasonable volume density fields).
- Why is a decoder MLP necessary when one can directly predict and volume render the class label logits from 3D convolution? I think this should be justified. Also, it is unclear what the hyperparameters in Table 5 exactly correspond to.
- Fig ?? is unreferenced in p12.

---

> ### Author Response · Authors · 2022-05-05
> **Responses**
>
> ### Q1 – ”new synthetic datasets are nice [...] but are [not to be] considered as a contribution”
> This is incorrect, Kubric had not claimed this dataset at submission time, and the only datasets that were claimed as contributions are tracked by [this discussion](https://github.com/google-research/kubric/discussions/206). As you can see the datasets from NeSF are not listed here, nor were claimed as contributions within the paper.
>
> ### Q2 – ”comparison against SparseConvNet [SCN] does not seem to be apple to apple”
> Recent work shows clear advantages on the use of NeRF density functions over point clouds; Fig. 2 in [Nerf Supervision](https://arxiv.org/pdf/2203.01913.pdf), motivating the investigation of inference tasks on density fields.
> We would like to emphasize that the comparison is unfair to NeSF, as SCN is provided with ground truth.
>
> While not included in the main paper, we have attempted to “reduce” SCN to work on point clouds extracted from the pre-trained NeRF model (i.e. your **proposal#1**), and SCN attains a performance comparable to NeSF in that preliminary experiment (mIOU 0.68 vs. 0.62, respectively). However, there are many heuristic decisions that had to be taken (applying a threshold over which to keep points, removing points above a certain z-level due to lack of sparsity within the scene floor, throwing away points that are not within a certain distance of a ground truth label). In addition to the unsatisfying heuristic decisions, we ended this line of comparison due to several additional factors uncovered: NeRF-derived geometry contains points internal to objects (not a surface point cloud) and these points will need to be labeled whereas NeSF will never receive this type of training signal, thresholding is unprincipled across scenes, sparseconvnet requires sparse point clouds and is not intended for points in interior of objects. Building such a baseline, we did not feel it would have correctly represented the performance of SCNs (however, we can include it in a revision if deemed necessary). That is why we preferred reporting SCN as an “oracle” upper bound on performance.
>
> Regarding **proposal#2**, training NeSF on GT 3D shapes; this requires quite a significant methodological addition as one will need to represent GT scenes with continuous density fields (composed of e.g. watertight meshes representing each object in a bounded volume hierarchy) on a TPU. Additionally, for Kubric unfortunately the generator does not yet expose such information in the API.
>
> Regarding **proposal#3** of using a differentiable rendering, this seems an interesting idea, but please note we are not trying to “beat” point cloud methods (e.g. beating SOTA), instead we are trying to shift the domain of analysis from point clouds to density fields.
>
> ### Q3 – "discussions should be made on Fig 4”
> For clarity, we assume you are referring to the low-data regime (10 scenes) the 2D & 3D mIoU in Fig.4, where 5 images (green) has a higher score than 10 images (purple). We had not discussed this extensively, as the **intended working regime for NeSF is on the opposite side of the spectrum** – when you can train on many scenes (easy to obtain), but only have few semantic annotations (expensive as they require manual intervention). With very limited number of scenes, **learning to generalize across scenes becomes difficult, overfitting more likely** (more images per scene will not necessarily help with generalization across scenes, though more scenes should), and we expect additional differences to be explained by the variance of random initialization.
>
> ### Q4 – ”discussions on the failure cases”
> Please note that we discussed failure cases **extensively**. However, the discussion is spread throughout the text, supplement, and supplemental video. We will consolidate the points in a subsection to aid in ease of reading if required.
>
> More specifically: sensitivity to reconstruction quality (Table 6) demonstrates the degradation in performance when a NeRF model quality degrades. We further discussed other limitations such as the inability to label objects with size smaller than the UNet grid (see: Sec 5.2 comparisons to baselines / quantitative comparisons / qualitative comparisons first paragraph, Sec 7 Conclusions and Limitations, Supplement B.1 qualitative results, Supplement B.2 model ablations), label leakage between geometrically adjacent objects (qualitative comparisons first paragraph), confusion between similar objects (B.4 Limitations), and loss in performance due to NeRF “floaters” (Supplement B.3 Multiview Consistency, Supplement Figure 13, Supplement B.4 Limitations, Video Timestamp 7:36).

---

> > ### Comment · Reviewer_kHpB · 2022-05-16
> > **Thanks for the responses**
> >
> > Dear authors,
> >
> > Thanks for the response. I am happy with the response for Q1 and 4-6. Regarding the rest:
> >
> > **Q2: Comparison against SparseConvNet.**
> > My point is that the experiments are not apple-to-apple by design, *regardless* of whether NeSF or SparseConvNet is advantageous, and thus the current comparison is pretty much meaningless. The baselines I proposed are meant more to bridge the discrepancy between NeSF and SparseConvNet. I am not expecting NeSF to (completely) beat point cloud methods or such, and I agree that analyzing segmentation with density field representations has its importance, but it's unclear why such baselines are presented in the first place. Therefore, I think at least one of the proposed experiments should be present. I can understand that functionalities of the Kubric dataset generator may be limited, but there are also other available datasets with 3D ground truth for this purpose [A]. So it doesn't seem like proposal #2 is undoable, putting aside the complexity of the datasets.
> >
> > **Q3: Fig 4.**
> > Apologies for not being clear. I meant that I would be expecting performance to be _strictly_ improving with the number of view supervision, but it does not seem to be the case looking the figure. I think this should be made clear.
> >
> > [A] Yu et al. "Unsupervised Discovery of Object Radiance Fields." ICLR 2022

---

> > > ### Author Response · Authors · 2022-05-18
> > > **Thanks for the questions**
> > >
> > > [updated to further clarify Fig.4]
> > >
> > > ### Q2: "Why such baselines (SparseConvNet) are presented in the first place"
> > > One of our core arguments is whether we could forego point cloud networks in favor of working on density, so we expected such a non-direct comparison to be asked by at least one of the reviewer. We were inspired by works in semi-supervised learning, where typically papers report a fully-supervised baseline as an (typically unreachable) golden standard.
> > >
> > > ### Q2: "At least one of the proposed experiments should be present"
> > > *As requested, we are happy to include a description of proposal1 to the supplementary in our revisions.*
> > > Regarding proposal#2, you are correct, it is not undoable, but it either requires very careful planning, or significantly reducing the complexity of the dataset to a few objects as in [A], which would limit our ability to study generalization. For the first option, note this would require computing 50k+ SDFs, and storing them while ensuring they are sufficiently precise and do not occupy too much memory/disk. We would expect this task to be as onerous as the post-processing of ShapeNet to a PBR-friendly format that is described in the Kubric paper?
> > >
> > > ### Q3: Fig 4
> > > Thanks for clarifying your question. And to make things even clearer, let us split the analysis along two axis:
> > > - with respect to the _number of scenes_ they are monotonically increasing. More specifically, look at bars of a single color, they are monotonic as you go 10→25→50→75→100, and this is true for any color, and regardless of whether you look at 2D/3D.
> > > - with respect to the _number of annotated semantic maps_ they are at times non-monotonic (and to be precise, this is not the number of views, as the number of views/scene is constant for NeRF training); however, the non-monotonicity disappears as we increase the number of training scenes (e.g. the group of bars for 3D w/ 25 semantic maps), and we believe the non-monotonicity to be _caused by overfitting_ in the few-scene regime.

---

> > > > ### Comment · Action_Editors · 2022-06-05
> > > > **Can you please include your final recommendation?**
> > > >
> > > > Hi kHpB ,
> > > >
> > > > Could you please add your official recommendation for this paper?
> > > >
> > > > Thanks,
> > > >
> > > > David

---

> ### Author Response · Authors · 2022-05-05
> **Responses (part2)**
>
> ### Q5 – “why is a decoder MLP necessary when one can directly predict and volume render”
> The execution of non-linear layers before/after a trilinear interpolation has been discussed extensively in the recent literature.
> While we discovered this aspect independently, executing a non-linear layer after, rather than before, the interpolation layers allows for higher frequency functions to be captured. Please see the discussions in Section 4 of Direct Voxel Grid Optimization, as well as Plenoxels.
>
> ### Q6 – “unclear what the hyperparameters in Table 5 exactly correspond to”
> The hyperparameters in Table 5 are for various parameters applied for the NeSF training training architecture (Figure 2) and respectively correspond to: whether random rotations are applied to the density grid during training (random rotations), the resolution of the density grid along the 3 spatial dimensions as referred to by epsilon in equation(4), the number of output features for each of 3 respective CNN layers of the [UNet](https://arxiv.org/pdf/1606.06650.pdf), and the respective number of hidden layers and hidden units per layer i.e. (number hidden layers, number hidden units) of the MLP (MLP). Thanks for pointing this out, we will include these descriptions in our revisions.

---

### Review · Reviewer_KQCV · 2022-05-05

**Summary Of Contributions:**

This method proposed a novel semantic understanding method. The approach takes posed images as input and constructs a neural radiance field as an intermediate representation. The 3D translation UNet is applied to convert the opacity field to a semantic field, and a neural rendering algorithm produces a rendered semantic mask. The semantic components are generalizable in novel scenes. Experiments are demonstrated on a novel synthetic dataset.



**Broader Impact Concerns:**

The paper has a well-written section discussing the potential broader impact concerns.

**Requested Changes:**

See the "Questions" and "Claims" parts above.

The reviewer will also appreciate the author's response to some of the comments above.

**Strengths And Weaknesses:**

Pros:
- Novel and technically sound approach.
- Writing is mostly clear and easy to parse. The structure of the paper is well organized, and the figures are clear and visually appealing.
- The semantic component can generalize to novel scenes.
- The geometry-only method demonstrates robustness to color shifts compared against the baseline etc.
- Thorough literature review and well-written limitation sections.

Cons:
- The method has several drawbacks compared to its closest related work, Semantic-Nerf (geometry only, modular pipeline, redundant representation).
- Experiments are only conducted in synthetic datasets. Overall it's hard to see the practical values.
- Both 2D and 3D baselines are not up-to-date. Semantic-Nerf used a weak segmentation network.
- The merits of the dataset are questionable.
- There are a few ambiguous claims on existing works and datasets.

The proposed geometry-only semantic field has a few drawbacks: 1. it requires two different representations at inference (one is Nerf representation, and the other is 3D voxel). 2. Error could be propagated along a modular pipeline — i.e., incorrect geometry from Nerf (likely to be noisy in practice as demonstrated in prior literature) will result in inaccurate 2D/3D semantics. 3. Appearance information is not well exploited.

The limitation section is well-written. That said, the submission would be significantly stronger if the author could at least resolve one or few of the discussed limitations, e.g., demonstrate real-world images or leverage appearance information provided by the neural radiance field.

I have some reservations about the merits of the proposed dataset.
This synthetic dataset is built upon basic object shapes and geometry primitives, and procedurally-generated object placements. It's hard to say how much benefit it could bring for advancing vision and graphics in practice. As a comparison, photorealistic rendering datasets, such as HyperSim and Synscapes, provide sensor photorealism while the underlying layout and assets are much more realistic than the proposed dataset. In addition, PBR rendering tools like BlenderProc and Clevr, provide users a reproducible script interface to generate a physical-based rendering dataset with similar quality but much better flexibility and controllability. The proposed dataset, however, is fixed.

Both DeepLab and SparseConvNet are quite old semantic segmentation networks published four years ago. It is highly recommended to use stronger 2D/3D baselines and a stronger 2D net for Semantic-Nerf.

Despite that robustness to color dataset shift demonstrated, relying on geometry makes it hard for NeSF to handle certain situations, e.g., semantically distinguishing a volleyball vs. a basketball or wall vs. painting.


Claims:
- "no RGB input is required for inference on novel scenes" this sentence could be misleading, considering building nerf requires RGB input.
- "Our method is the first to learn semantics from the geometry stored within a 3D neural field representation." I would be careful about this claim as it could be misleading. As the author mentioned, The pseudo-labeled version of Semantic-Nerf requires only posed RGB images as well. The semantics and geometry for semantic-Nerf are jointly learned under a unified representation.
- "they lack the scale, detail, and precision necessary to simultaneously evaluate 2D and 3D semantic segmentation": I have some reservations about the claims on existing datasets. ScanNet and Matterport3D have been widely adopted and used in common benchmarks. Both are large, captured in the real world, and can provide accurate 2D and 3D semantic benchmarks.
- "NeSF is trained using purely 2D signals, and can be trained with as few as one labeled image per scene" — this could be the drawback since semantic nerf only requires 2D annotated images without building a nerf during training. Training images could be single-shot instead of a multi-view collection.

Questions:
- Have you tried to leverage the radiance field into the UNet input as well? This should be fairly straightforward.
- How about training SparseConvNet on the 3D voxelized occupancy field extracted from Nerf or dense point cloud produced by a SFM-MVS pipeline. Either way could provide a fair comparison.
- What will be the downstream applications? Do you anticipate such a method could replace semantic fusion for visual scenes in the future?
- Do you anticipate the method will work well for real-world datasets such as ScanNet?

Minor:

Another interesting work worth discussion (not required as it's public post the submission):
[1] Fu et al. Panoptic NeRF: 3D-to-2D Label Transfer for Panoptic Urban Scene Segmentation, arXiv 2022.

---

> ### Author Response · Authors · 2022-05-06
> **Responses**
>
> Thanks for the thoughtful review. It seems to us that the review is focused on comparing NeRF to SemanticNerf and/or SparseConvNets, but we believe this sidesteps the point of our fundamental contribution. While NeRF and SemanticNerf target the same objective, they **investigate orthogonal areas of research** – let us explain in detail:
>
> SemanticNerf obtains generalization from a **pre-trained** 2D semantic segmentation backbone. That is, generalization is achieved by identifying patterns in **appearance space**, after which NeRF is used to perform 3D fusion.
> Conversely, NeSF obtains generalization by identifying patterns in **geometric space**, by recognizing the 3D structure of objects irrespective of their appearance.
>
> Clearly, these two lines of research could be combined, but in our research we *attempted to investigate a new problem, rather than building a system*. This can be further understood by considering that SemanticNerf had not yet appeared on arXiv by the time we started this investigation.
>
> A similar objection can be made with respect to SparseConvNets. We are not attempting to “beat” these architectures, as our problem is not formulated on point clouds, and we do not claim to be better (after all point cloud architectures have been investigated for 5+ years, while this is the first attempt at working with NeRF densities). Instead, we include SparseConvNet as an benchmark of upper bound on performance, while the fundamental questions we are asking are:
>
> - “is there any semantic information in NeRF density fields, and what would be an architecture to extract such information?”
> - ”is there a more direct way to perform such a task, as opposed to first extracting point clouds from NeRF densities (an operation we found unprincipled, ill-suited to SparseConvNet, and lossy)?”
>
> We are of course more than happy to answer any of your questions or concerns in detail, but before we do that, **we would kindly ask you to carefully consider the TMLR review guidelines**, as they are significantly different from CVPR/SIGGRAPH/NeurIPS/etc. In particular, please refer to the [Acceptance based on claims](https://medium.com/@hugo_larochelle_65309/announcing-the-transactions-on-machine-learning-research-3ea6101c936f#:~:text=Acceptance%20based) subsection of the TMLR announcement, as well as the [TMLR Evaluation criteria](https://www.jmlr.org/tmlr/editorial-policies.html#:~:text=evaluation%2520criteria%2520and%2520certifications).
>
> With this in mind, the core claim we aim to verify is **“can 3D semantics be distilled solely from 3D NeRF density fields”**, and we believe our analysis has validated this claim. We would also like to emphasise that TMLR is a Machine Learning rather than a Computer Vision venue, and that in ML venues the use of synthetic data as the primary evaluation benchmark is ubiquitous (e.g. any paper based on CLEVR).
>
> We would like to conclude with a rebuttal of two observations from the initial review:
>
> - ”The proposed dataset, however, is fixed”: this is incorrect. We provide the Kubric script that generated this dataset, which is a dataset contribution of our work, hence allowing future work to build more challenging settings.
>
> - ”ScanNet and MatterPort3D have been widely adopted and used in common benchmarks”: while this is true, trivially applying NeRF to such datasets produces low-quality NeRF reconstructions, and the development of NeRF models for 3D data in the wild is completely orthogonal from the investigation of semantic analysis on NeRF models.
>
> We are happy to provide detailed responses to questions, but we would first kindly ask you to carefully consider the TMLR guidelines, as some comments are not well aligned with the unique review style dictated by TMLR.

---

> > ### Comment · Reviewer_KQCV · 2022-05-06
> > **Re:**
> >
> > Thanks for the reply. The reviewer has reread and acknowledged the TMLR reviewing guideline and the motivation behind it.
> >
> > As mentioned in the initial review, the reviewer appreciates the novel, technically sound approach and believes such exploration is new and has the advantage of generalizing. The reviewer also likes the paper presentation and discussions. In particular, the author's extensive discussions of its limitations are greatly appreciated.
> >
> > The reviewer is convinced the paper's core claim is to verify the ability for "semantic understanding from density field of NeRF." That said, just as with every new research question we would like to answer, it is essential to ask a scientific question "is it worth exploring this problem?"
> >
> > Hence, the reviewer believes it is fair for the paper to convince the audience whether extracting semantics merely from NerF geometry is a plausible direction for the community to explore, given that 1) appearance is readily available in NeRF, which is required to compute anyways, and 2) NeRF geometry is noisy, in particular in complex real-world scenes.
> >
> > Therefore, the reviewer believes many questions/comments (some are shared with other reviewers) are valid questions to consider answering.
> >
> > Let me give a few examples to justify my comments:
> > - As mentioned in the paper, SparseConvNet is conducted with GT geometry, an oracle, which is *not fair* to the proposed method. The reviewer wants to point out it would be fairer to run SparseConvNet using either noisy NerF geometry or the MVS geometry.
> > - Even for ML journals and conferences, real-world experiments are commonly well appreciated. And NeRF's density field tends to become worse in the real world than synthetic ones. I don't think it's unreasonable to ask for a discussion on how well the author anticipates NeSF will work in the real-world dataset. This will serve the goal of verifying the core claim better -- readers will know to what extent it works in a less satisfactory setting. It is perfectly fine to just answer this question by claiming Nerf cannot provide a good density field in ScanNet -- not the reason to reject the paper as improving Nerf geometry is not the goal here.
> > - It is essential to avoid some potentially misleading claims. For this reason, I pointed out some in the paper with justifications. E.g., claims on existing real-world datasets could be really misleading.
> >
> > The reviewer wants to reemphasize that, as a reviewer, I genuinely wish to help make every submission stronger. The initial review raised several questionable claims in papers and questions that I believe could be nice to address. The purpose of asking these questions is not to find excuses to reject this paper. Considering the short TMLR reviewing cycle and all the reviews are subjective, it is up to the author to decide what questions to address.

---

> > > ### Author Response · Authors · 2022-05-09
> > > **Re:**
> > >
> > > Thank you very much for your reply, and for acknowledging the validity of our work.
> > >
> > > ### Q1 – "Method could replace semantic fusion for visual scenes in the future?" and "plausible direction for the community to explore"
> > > Given the proliferation of NeRF research, we envision that we are not far from completely replacing depth/lidar sensors with efficiently trained NeRF model at large scale. This motivates the question of whether it is possible to design semantic/panoptic segmentation techniques that *operate directly on such representations*, without the need to access the original images to train the model, nor extracting a point cloud.
> > >
> > > Regarding the “noise in density function”, please note that this is not an issue if the scene is acquired carefully, and that recently researchers have started the investigation of suitable regularizers (e.g. see[RegNerf](https://m-niemeyer.github.io/regnerf/index.html) or [RefNerf](https://arxiv.org/abs/2112.03907)).
> > >
> > > ### Q2 – ”merely from NerF geometry [...] given that appearance is readily available in NeRF”
> > > We could add an ablation experiment where the predicted color is added to the MLP that predicts semantics, if necessary. However, note that color in a NeRF model is a view-dependent property, hence a non-trivial way of *comparing view-dependent radiance distributions* would become necessary. Hence, we expect the use of pre-trained 2D features a-la SemanticNerf to be more effective, but this would be a system paper combining two techniques.
> > >
> > > ### Q3 – SparseConvNet using noisy NerF geometry
> > > As discussed in *Reviewer#kHpB Q2*, we have attempted a less unfair comparison resulting in comparable semantic segmentation performance, whose description could be added to the supplementary, if requested.
> > >
> > > ### Q4 – SparseConvNet using noisy the MVS geometry
> > > As for attempting to use MVS geometry, we have attempted to use COLMAP to extract dense point clouds (and originally intended to include that comparison) but the results were of very low quality, with either the pipeline failing to converge altogether, or the point cloud containing significant (so called) "edge fattening" (which would have lead to extremely poor semantic predictions).
> > >
> > > ### Q5 – On real-world dataset
> > > We anticipate that in regions where novel view synthesis succeeds, so will NeSF. In this sense, the recent developments of NeRF techniques targeting medium (mipNerf360, Urban Radiance Fields) or large (BlockNerf) scenes mean that such experiments will become possible relatively soon.
> > >
> > > ### Q6 – Nerf cannot provide a good density field in ScanNet
> > > Indeed, it is exactly as you say. We believe the biggest issue is the presence of large amounts of motion blur and changes in exposure. There are some early attempts to address these (DeblurNerf), but these have only been tested on a few hand-picked scenes. Once NeRF training problems on ScanNet are resolved, we believe our method will perform effectively.
> > >
> > > ### Q7 – It is essential to avoid some potentially misleading claims
> > > We have no issue with carefully revising these statements so to avoid any misunderstandings.
> > >
> > > Please let us know if you have any further questions.

---

> > > > ### Author Response · Authors · 2022-05-11
> > > > **Request for clarification**
> > > >
> > > > We would also like to request clarification on the following note (claims section):
> > > >
> > > > * "NeSF is trained using purely 2D signals, and can be trained with as few as one labeled image per scene" — this could be the drawback since semantic nerf only requires 2D annotated images without building a nerf during training. Training images could be single-shot instead of a multi-view collection.
> > > >
> > > > Could the reviewer please further clarify their concern? SemanticNeRF indeed requires building a NeRF like model during training (from multiple RGB views).

---

> > > > > ### Comment · Reviewer_KQCV · 2022-05-24
> > > > > **RE**
> > > > >
> > > > > There might be some confusion in my comments.
> > > > >
> > > > > Firstly, let me clarify the two ``training'' stages: 1) per-scene training, which essentially fits SemanticNeRF / NeSF given image collections from a new scene; 2) offline training, which trains the 2D semantic seg / 3D semantic understanding from an offline training dataset.
> > > > >
> > > > > What I meant is that the different requirement for offline training dataset in the two settings needs to be mentioned:
> > > > > - For offline training semantic components, NeSF needs one or few image-semantic pairs; the annotated images must capture from a multi-view collection of a scene -- the multi-view capturing is required due to the requirement of NeRF building during the offline training.
> > > > > - In contrast, SemanticNerf only requires single-view 2D semantic annotations for its semantic components, just like a standard segmentation dataset without the need for labeled images from a multi-view image scene capturing. All photos could come from a single shot setting, like ADE20k or COCO.

---

> > > > > > ### Author Response · Authors · 2022-05-26
> > > > > > **RE**
> > > > > >
> > > > > > Yes, what you write is correct.
> > > > > > We will try out best to find a way to formalize this (perhaps with the help of algebra?), and integrate this in our revisions.
> > > > > > Likely the related works section seems the most appropriate place to do so, don't you think?

---

> > > > ### Comment · Reviewer_KQCV · 2022-05-24
> > > > **Thanks for the reply.**
> > > >
> > > > Q1: "Given the proliferation of NeRF research, we envision that we are not far from completely replacing depth/lidar sensors with efficiently trained NeRF model at large scale." -- I would be very cautious about this claim (and Q5/Q6) as it might take years -- there is no evidence that a monocular NerF could perform closely as well as depth sensor/lidar on real-world ScanNet, KITTI, etc., or is close to state-of-the-art MVS on a well-controlled multi-view benchmark, e.g., Tanks and Temple. Neural surface modeling seems to be closer to the goal than neural radiance volumes. Besides, I would be cautious about the conclusions drawn from synthetic data.
> > > >
> > > > Q2: A simple ablation that takes the second last feature layer or radiance along the surface normal direction should suffice.
> > > >
> > > > Q3 / Q4: Thanks. And I believe describing your efforts and showing how MVS + 3D seg do could greatly help answer the question "why do we want to extract semantics from nerf". So I believe it's relevant. That said, make sure you leverage the full colmap pipeline including depth fusion and triangulation.
> > > >
> > > > Q7: Thanks. Please revise accordingly in the final version.

---

> > > > > ### Author Response · Authors · 2022-05-26
> > > > > **RE**
> > > > >
> > > > > ### Q1: I would be very cautious about this claim (and Q5/Q6) as it might take years
> > > > >
> > > > > Actually, there are already strong signals about its progress. Not only generally one of the most admired features of NeRF models are depth maps, but their quality has recently been certified on public leaderboards for "in-the-wild" data: http://www.cvlibs.net/datasets/kitti-360/leaderboard_nvs.php?task=nvs_rgb
> > > > >
> > > > > Clearly, this is only true for the "many images/scene" setting, while for "few images/scene" settings classical methods are still superior (e.g. depth estimates from stereo), but this is not the application scenario we are considering in this paper.
> > > > >
> > > > >
> > > > > ### Q2:  A simple ablation that takes the second last feature...
> > > > > Yes, think this is feasible, and will be added to our revisions.
> > > > >
> > > > > ### Q3 / Q4:  MVS + 3D
> > > > > Sounds good, we will add the experiment description to our supplementary, as well as include images of the generated COLMAP dense point clouds (after verifying depth fusion/triangulation were used).

---

> > > ### Comment · Action_Editors · 2022-05-13
> > > **Thanks for participating in the discussion, any updates?**
> > >
> > > Hi KQCV,
> > >
> > > Thanks for the review and your clarifications on your intentions on the questions. I wanted to ask whether you had any updated thoughts given the authors' responses below. In particular, the authors have presented a few options of things they could do (e.g., adding an ablation involving predicted color, the SparseConvNet experiment that could be added to the supplemental) and they have asked a clarification question on one of the notes in the claims section. Sorry for prodding, given that discussion's already happening (thank you!), but the TMLR timeline's fairly short.
> > >
> > > Thanks,
> > >
> > > David

---

### Decision · Action_Editors · 2022-06-10

**Recommendation:** Accept with minor revision

**Comment:**

After an extensive discussion between the reviewers and the authors, all three expert reviewers are in favor of acceptance. The editor agrees with the reviewers and is also in favor of acceptance. The discussion surfaced a few small changes that the reviewers and authors (at least in the editor's view) agreed upon.Here are the specific changes the editor sees as needed (and who asked for them):
1.  (kHpB, KCQV) Please include in the supplementary the SparseConvNet using noisy Nerf Geometry baseline discussed in kHpB Q2 (proposal #1 - the one that has already been completed; no need to do any new ones). This was also the one agreed upon with KCQV in Q3 .
2. (KQCV) Please address KQCV's Q2 request for a simple ablation (i.e., one that takes the second-to-last feature layer).
3. (kHpB) Please add to Figure 4's caption to help readers understand it better. An abbreviated version of what was written in the response to kHpB on 17 May is perfect.
4. (u1Zd) Please add a few short sentences to address the discussion regarding u1Zd Q1 ("nerf took us away from the clutches of synthetic data"). The editor agrees with u1Zd that a simple demo could be nice, but also not necessary
5. (u1Zd) Please add commentary about data augmentation (Q4) and new semantic classes (Q6)

Based on the discussion between the reviewers and the authors, the editor has the following suggestions that are entirely up to the authors but may improve the final version of the paper:
1. The discussion with KQCV about the claim of the method replacing semantic fusion might be of interest to readers. If the authors can somehow integrate some of this discussion, it might be of interest to readers. If there is not a good spot, that is also fine.
2. A consolidated discussion of failure cases could be helpful. The paper does discuss failures throughout the paper. However, a short consolidated sentence or two flagged with "Failure cases" might help readers who are reading the paper more casually.

The editor also would like to thank the authors and reviewers for engaging in discussion productively throughout this process.

---

> ### Author Response · Authors · 2022-06-18
> **Thank you to editor and reviewers, and time estimate for revision.**
>
> Thanks to the editor and reviewers for their support of our work! We greatly appreciate your engagement in a thorough discussion, and the feedback you have provided. We acknowledge the set of requested changes and will provide a revised version for verification shortly.